# The accelerating loss and shifting dynamics of US tidal wetlands

Xiucheng Yang [1] ✉, Shi Qiu [1], Kevin D. Kroeger[2], Zhiliang Zhu[3], Scott Covington[4], Nicholas J. Murray [5] & Zhe Zhu [1] ✉

Tidal wetlands are critical ecosystems for coastal sustainability, yet despite growing regulatory protection, they continue to decline globally. Their long-term resilience to interacting chronic stressors and extreme events remains uncertain, in part because comprehensive, high-frequency monitoring has been lacking. While direct land-use conversion has been substantially restricted in the United States, the true trajectory of these protected habitats has remained unclear. Here, we use four decades of high-resolution satellite records to analyze the shifting dynamics of US tidal wetlands. We reveal a widespread and previously unquantified acceleration in the rate of tidal wetland loss, amounting to a net loss of −1640 km² at the rate of −40.53 km² year⁻¹, accelerating by −0.73 km² year⁻², of which tidal marsh contributed the majority of this loss with a cumulative decline of 1567 km². Furthermore, we show that the drivers of this decline are shifting: while chronic stressors like relative sea level rise have caused the largest cumulative loss (~60% of the total area loss), acute shocks from extreme weather now dominate (1.4 times that of the chronic stressors) the acceleration of that loss. By contrast, direct human activities were a minor driver, accounting for only 4% of total observed losses. These findings indicate that the resilience of these protected ecosystems is declining. It provides an urgent warning that existing conservation strategies, initially concerned with direct human impacts and increasingly focused on relative sea level rise as a slow-moving pressure, are ill-equipped for a future of increasing extreme weather events and highlights the need to redesign adaptation policies.

Tidal wetlands, including tidal marsh, mangrove, and tidal flats[1], are dynamic ecosystems shaped by regular and irregular tidal fluctuations within the intertidal zone. Tidal wetlands render unique ecosystem services of notable worth relative to terrestrial environments[2], including biodiversity, water quality maintenance, habitat provision, flood control and storm surge mitigation, shoreline protection, and carbon sequestration. On an area-normalized basis, global assessments rank tidal wetlands as the second most valuable ecosystem after coral reefs, largely because their services are delivered in densely populated coastal zones[3], where they directly reduce risk and support human livelihoods[4,5]. Yet, these vital habitats have undergone significant degradation and decline due to intensive human activities. For example, since European colonization in the United States (US), extensive conversion of tidal wetlands has occurred, primarily through land-use practices involving wetland filling and drainage associated with agriculture, urbanization, and infrastructure development[6]. In

[1]Department of Natural Resources and the Environment, University of Connecticut, Storrs, CT, USA. [2]Silvestrum Climate Associates, Woods Hole, MA, USA. [3]Formerly with the United States Geological Survey, Reston, VA, USA. [4]U.S. Fish and Wildlife Service, Vancouver, WA, USA. [5]College of Science and Engineering, James Cook University, Townsville, QLD, Australia. ✉e-mail: xiucheng.yang@uconn.edu; zhe@uconn.edu

addition to direct conversions into intensive land-use types, tidal wetlands also face threats from both landward (e.g., reduced sediment loads, altered hydrology, and excess nitrogen loading) and seaward processes (e.g., accelerating relative sea level rise and extreme weather events)[7–10]. With such a broad range of spatially and temporally varying change processes affecting tidal wetlands, little is known about the natural adaptation of tidal wetlands to these change drivers, where direct conversions are relatively infrequent in modern times.

Despite growing recognition of their ecological and climatic importance, tidal wetlands still experienced substantial global declines in extent and condition over recent decades, primarily driven by intense human activities and increasing climate-related stressors[1,8,11,12]. Recent global syntheses estimate a net loss of ~4000 km² of tidal wetlands since the late twentieth century[1], including ~1400 km² of tidal marsh[8,13], underscoring the continued vulnerability of these ecosystems worldwide despite expanding conservation frameworks. The adaptability of tidal wetlands to climate change, when direct human alterations and land use change are restricted, remains an enigma. Concurrently, the effectiveness of societal conservation of these ecosystems achieved through regulatory protection has been regarded as a topic of significant importance. The US is an ideal setting for investigating tidal wetland resilience and adaptation to climate change. Over the past 40 years, the US has implemented regulatory policies specifically designed to safeguard tidal wetlands by limiting direct land-use conversion[7,14]. Hence, it is intriguing to consider the US as a case study to investigate the outcomes of extensive tidal wetland protection measures as a counterbalance to the accelerating pace of climate change and the associated degradation of habitats and biodiversity.

Historically, tidal wetland extent has been difficult to evaluate synoptically due to the inherent dynamic and heterogeneous nature of coastal systems[1]. Water level fluctuations lead to varying inundations of tidal wetlands over time and inconsistent spectral values in satellite images. That variability has been a significant barrier for continuous monitoring of tidal wetlands. Traditionally, analysis has been limited to optimal satellite images, such as those captured during similar high tide or low tide conditions, have been selected based on tide modeling and measurements from gauge stations[12,15,16]. Previous large-scale assessments have been limited to infrequent, epoch-based time slices that under-sample the full variability of these dynamic systems (refer to Supplementary Fig. S1)[7,8,17]. These methods are largely blind to the crucial, inter-annual rates of change and acceleration that define an ecosystem's true trajectory and vulnerability, leaving a critical gap in our understanding of whether decades-old conservation policies remain effective under accelerating sea-level rise and increasing climate volatility.

Here, we provide an observation-based assessment of long-term tidal wetland trajectories across the conterminous US. We developed a comprehensive monitoring framework that leverages the full 39-year Landsat 4–8 archive (176,223 images) to produce the annual, 30-m resolution map series of tidal wetland extent and change for the conterminous US (1985–2023). Our approach uses a dense time series analysis[18] designed to overcome previous monitoring limitations by explicitly accounting for tidal variability. This high-frequency time series of the US coastline allows us to move beyond static maps to continuous tracking, enabling continental-scale quantification of rates, acceleration, and shifting drivers of tidal wetland change.

## Results
### A four-decade view of accelerating tidal wetland loss
Our annual map series reveals that despite decades of regulatory protection, the loss of US tidal wetlands is not only persistent but is accelerating. In 2023, the total estimated area of tidal wetlands in the conterminous US was 19,931 [±1046] km² [95% confidence intervals]. The net loss from 1985 to 2023 was −1640 [±72] km², or 7.60 [7.54,

7.66] % of the area in 1985 (Table 1), an area more than twice the size of Singapore. Tidal wetlands underwent an annual net loss of −40.53 [−43.40, −37.76] km² year⁻¹. More importantly, the loss of tidal wetland accelerated at a rate of −0.73 [−1.55, −0.06] km² year⁻². This high-frequency, continuous analysis enables direct quantification of tidal wetland dynamics associated with climate-related drivers, such as sea level rise and extreme weather events (Fig. 1d–i).

In 2023, tidal wetlands in the conterminous US consisted of 16,119 [±675] km² tidal marsh, 2434 [±325] km² mangrove, and 1378 [±46] km² tidal flats. Tidal marsh contributed to the preponderance of loss, declining at an annual loss trend of −39.13 [−41.44, −37.69] km² year⁻¹ (Fig. 2a), with a cumulative loss of almost −1567 [−1492 to −1632] km² (8.86%) from 1985 to 2023. Mangrove areas (Fig. 2b) remained at similar levels between the years 1985 and 2023. The extent of mangroves shows substantial dynamics but no significant gain or loss during this period. Tidal flats experienced a slight cumulative decrease of −58 km² based on an annual loss trend of −0.70 [−1.03, −0.38] km² year⁻¹.

At a subbasin hydrological unit scale (213 units that contain tidal wetlands) (Fig. 1a), 153 subbasin units (72%) experienced tidal wetland loss trends, and the loss trend in 136 subbasins (64%) is accelerating. By comparison, only 44 subbasin units (one-fifth) experienced tidal wetland gains, and 11 units (5%) have a trend of increasing gains. The cumulative net change parsed by longitude (Fig. 1b) and latitude (Fig. 1c) highlights the dominance of Gulf coast marsh loss, with the Mississippi River Delta Plain contributing to the largest single regional deficit.

Tidal wetlands along the Pacific coast generally exhibited net gains, while extensive and persistent losses dominated the Gulf and Atlantic coasts. For example, in San Francisco Bay (Fig. 1d), large-scale restoration projects since the late 1990s have produced substantial tidal marsh recovery. The northern Gulf, particularly the Mississippi River Delta Plain (Fig. 1e), depicted the most pronounced acceleration of extensive loss. Delaware Bay (Fig. 1f) exhibited seaward edge erosion along marsh margins, while northern Florida (Fig. 1g) revealed marsh-to-mangrove conversion through gradual encroachment. Mangrove systems displayed both dieback and expansion: dieback in Florida was linked to extreme weather events (Fig. 1h), while poleward encroachment into tidal marshes was evident in northern Florida (Fig. 1i). Together, these patterns demonstrate a coupled system of tidal marsh retreat and mangrove migration under climate forcing and disturbance regimes.

Our maps that integrate both loss/gain rates and acceleration/deceleration trends highlight hotspots of shifting tidal wetland dynamics. Along the Gulf coast, particularly in Louisiana and Florida, areas that historically experienced the most rapid wetland loss now exhibit signs of deceleration, likely reflecting the impact of large-scale restoration efforts. In contrast, sections of the Atlantic coast, where the long-term resilience of tidal wetlands remains uncertain, show comparatively low absolute rates of loss but are marked by accelerating trajectories of loss, underscoring heightened vulnerability.

The annual analysis revealed distinct patterns of tidal wetland cover change trajectories (Fig. 2c). The annual tidal wetland loss outweighs the annual gain for most of the years, and the spikes in annual tidal wetland losses throughout the record echo the recurrences of extreme weather events, mainly including exceptional drought in Louisiana linked to the 1987–88 La Niña event[19], the freeze event in Florida during the winter of 1989 to 1990[20], Hurricane Andrew in Florida in 1992[21], El Niño in 1997–1998[22], Hurricane Wilma/Rita/Katrina in the Gulf of Mexico in 2005[23–25], Hurricane Gustav in 2008 with compound impact after the hurricanes in 2005[26], another hard winter freeze in Florida in January 2010[27], Hurricane Irma in 2017[28], and Hurricanes Laura and Ida in Louisiana in 2020 and 2021. Our maps show that following these acute events, impacted wetlands recovered at varying rates, but this recovery was often insufficient to prevent

**Table 1 | Conterminous US tidal wetland extent in 1985 and 2023, change trend, and strength of trend [95% confidence intervals] for three coasts**

| | | Tidal marsh | Mangrove | Tidal flats | Total |
|---|---|---|---|---|---|
| Pacific coast | Extent 1985 (km²) | 277 [±12] | Not applicable | 471 [±16] | 748 [±28] |
| | Extent 2023 (km²) | 308 [±13] | Not applicable | 473 [±16] | 781 [±29] |
| | Change trend (km² year⁻¹) | 1.23 [1.13, 1.31] | Not applicable | 0.20 [0.09, 0.27] | 1.40 [1.22, 1.50] |
| | Strength of trend (km² year⁻²) | Not significant | Not applicable | 0.02 [0.00, 0.03] | Not significant |
| Atlantic coast | Extent 1985 (km²) | 7195 [±301] | 81 [±11] | 411 [±14] | 7687 [±326] |
| | Extent 2023 (km²) | 7061 [±296] | 153 [±20] | 347 [±12] | 7561 [±328] |
| | Change trend (km² year⁻¹) | −2.75 [−3.54, −2.11] | 1.67 [1.50, 1.84] | −1.83 [−1.90, −1.78] | −3.35 [−3.93, −2.62] |
| | Strength of trend (km² year⁻²) | −0.37 [−0.44, −0.29] | Not significant | Not significant | −0.40 [−0.44, −0.36] |
| Gulf of Mexico | Extent 1985 (km²) | 10,214 [±428] | 2384 [±318] | 538 [±18] | 13,136 [±764] |
| | Extent 2023 (km²) | 8749 [±366] | 2281 [±304] | 559 [±19] | 11,589 [±689] |
| | Change trend (km² year⁻¹) | −37.65 [−39.74, −35.64] | −1.33 [−2.43, −0.05] | 0.67 [0.38, 1.03] | −38.92 [−41.49, −36.02] |
| | Strength of trend (km² year⁻²) | Not significant | Not significant | −0.23 [−0.25, −0.19] | Not significant |
| Conterminous United States | Extent 1985 (km²) | 17,686 [±740] | 2464 [±329] | 1420 [±48] | 21,571 [±1118] |
| | Extent 2023 (km²) | 16,119 [±675] | 2434 [±325] | 1378 [±46] | 19,931 [±1046] |
| | Change trend (km² year⁻¹) | −39.13 [−41.44, −37.69] | Not significant | −0.70 [−1.03, −0.38] | −40.53 [−43.40, −37.76] |
| | Strength of trend (km² year⁻²) | −0.58 [−0.81, −0.27] | Not significant | −0.19 [−0.21, −0.16] | −0.73 [−1.55, −0.06] |

Temporal trends were estimated using Sen's slope and assessed for significance using the Mann–Kendall test at the 95% confidence interval. Negative values of the change trend indicate a loss of tidal wetlands, whereas positive values indicate a gain. Negative values of strength of trend indicate decelerated gain or accelerated loss (see Supplementary Table S1 for an illustration of the acceleration and deceleration). Mangroves are not present along the US Pacific coast.

permanent, cumulative loss, indicating a clear decline in ecosystem resilience.

To understand the forces behind this accelerated decline, we conducted a comprehensive driver analysis based on the manual interpretation of a stratified random sample (see "Methods"). Our sample-based analysis confirms that chronic stressors (primarily relative sea level rise) are the largest overall driver of cumulative permanent loss, accounting for 59% of the total area lost (Fig. 2d and Supplementary Table S2). However, when we analyzed the drivers of the acceleration of this loss, our sample-based estimates revealed a critical reversal: the acute shocks of extreme weather events have emerged as the dominant force, with a contribution to the accelerating loss rate 1.4 times that of chronic stressors (Supplementary Table S2).

## Widespread and accelerating decline of tidal marsh

Tidal marsh, which constitutes approximately 80% of the US tidal wetland area, is the primary source of national-scale net loss. From 1985 to 2023, tidal marsh area declined by 1567 km² (Fig. 2a). This loss is driven by a combination of drivers. Chronic stressors were the largest contributor to permanent marsh loss (68%) and fluctuations (18%) (Supplementary Table S3). This is evident in the widespread failure of marshes to adapt to relative sea level rise through landward migration. Our analysis of marsh area across elevation gradients (Fig. 3) reveals that while most loss occurred at low elevations, there were no corresponding significant gains at higher elevations (MK test with 95% confidence interval), indicating that lateral migration is not keeping pace with seaward marsh loss. In addition to physical barriers, such as roads and developed uplands, forested areas adjacent to marshes may exhibit resistance to early stages of sea level rise, thereby delaying landward marsh migration and ecological transition[29–33]. We note that this analysis uses a static DEM from the 2000–2015 period as a consistent spatial reference to assess lateral migration and does not account for dynamic vertical processes, such as accretion or subsidence. The impact of chronic stressors varies regionally, with the

most severe cumulative decline (14% since 1985) occurring in the Gulf of Mexico, while the trend of loss is now accelerating most significantly (MK test with 95% confidence interval) along the Atlantic coast (Table 1 and Supplementary Fig. S2).

While chronic stressors drive the largest cumulative loss of marsh, extreme weather events are critical agents of large-scale, abrupt loss and are a significant factor in the acceleration of the decline. Our high-frequency analysis reveals the compounded impacts of sequential hurricanes, which are often missed by other studies relying on sparser satellite records or lacking the capacity to detect such rapid changes[7,8]. A prime example occurred between 2005 and 2008 in Louisiana, where Hurricanes Katrina, Rita, and Gustav caused a total, compounded decline of 388.7 ± 16.3 km² of tidal marsh before the system could recover (Supplementary Figs. S7 and S10).

## Dynamic balance and hidden vulnerability of mangroves

In contrast to tidal marshes, the total area of US mangroves remained stable from 1985 to 2023 (Fig. 2b and Table 1). However, this net stability masks two opposing dynamics that reveal hidden vulnerability.

First, mangroves are successfully adapting to chronic warming through rapid, climate-driven poleward migration. Our annual maps quantify a significant acceleration in this migration at a rate of 4.05 ± 0.16 km² year⁻¹ north of 26°N (Fig. 4). This expansion into areas formerly occupied by tidal marsh or freshwater wetlands is so substantial that it is the primary driver of permanent mangrove gains (Supplementary Table S3).

Second, this expansion is almost perfectly offset by severe, permanent losses caused by extreme weather events, such as hurricanes and freezing winters, which are the dominant driver of mangrove loss and fluctuations (Supplementary Table S3). A critical innovation of our high-frequency approach is the ability to distinguish healthy mangrove ecosystems from a distinct condition class we term mangrove dieback, characterized by widespread leaf and branch loss that persists for more than a growing season[34]. This class, which cannot be identified by lower-frequency monitoring, serves as a direct, spatially explicit indicator of

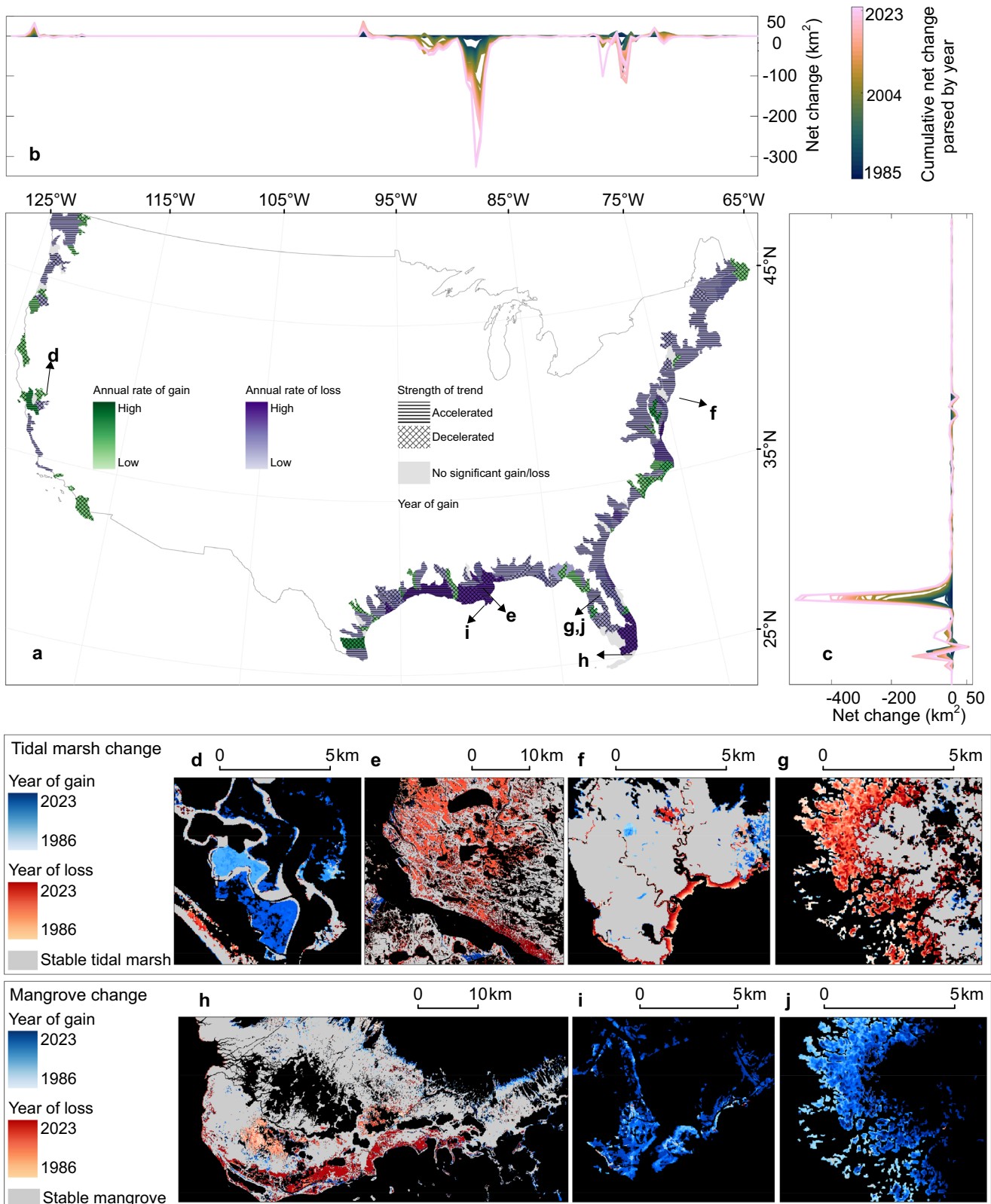

declining ecosystem resilience. Our analysis reveals a critical increase in the occurrence of large-scale mangrove diebacks following major hurricanes and severe freeze winters, particularly in the historical core of their range in South Florida. Hurricane-related dieback (notably in 2005 and 2017) is concentrated along seaward mangrove margins, whereas dieback following extreme freeze events (notably in 1990 and 2010) occurs predominantly along upland boundaries. For instance, our

analysis of Hurricane Irma in 2017 revealed that 63% of the 210 km² of dieback had failed to recover by 2021. This persistent, catastrophic damage, which signals a loss of ecosystem resilience, is largely invisible to lower-frequency global monitoring products that reported stability or even gains in the same period[1,11]. These hotspots of dieback in the Everglades signal ecosystems on the brink of collapse, warranting urgent restoration interventions[28,35].

**Fig. 1 | Maps of tidal wetland change from 1985 to 2023 in the conterminous US.**
**a** Occurrence of color cells, green, purple, and black, indicates the annual trend of loss, gain, and no significant change, respectively, at subbasin hydrological units during the 1985–2023 period. The horizontal or crossing stripes present the accelerated or decelerated trend, and when the symbols for accelerated and decelerated are absent, the strength of the trend is not significant. Temporal trends were estimated using Sen's slope and assessed for significance using the Mann-Kendall test at the 95% confidence interval. **b**, **c**, Cumulative net change amounts from 1985 to 2023 along the longitude (**b**) and latitude directions (**c**) respectively. **d-g** Hotspots of tidal marsh cover change, including restoration in San Francisco

Bay (38.14° N, 122.29° W) (**d**) extreme losses in the Mississippi River Delta Plain (29.74° N, 89.85° W) (**e**) seaward edge erosion in Delaware Bay (39.20° N, 75.12° W) (**f**) and gradual tidal marsh decline due to mangrove encroachment in north Florida (28.85° N, 82.71° W) (**g**). **h–j** Mangrove cover changes, including dieback owing to extreme weather events in Florida (25.15° N, 81.07° W) (**h**), and encroachment into tidal marsh in the Mississippi River Delta Plain (29.11° N, 90.22° W) (**i**) and) at the same location as (**g**) shown from the perspective of mangrove migration (**j**). The US boundary is from the US Census Bureau. Refer to the whole online interactive map for details at https://gers.users.earthengine.app/view/tidalwetlandcover.

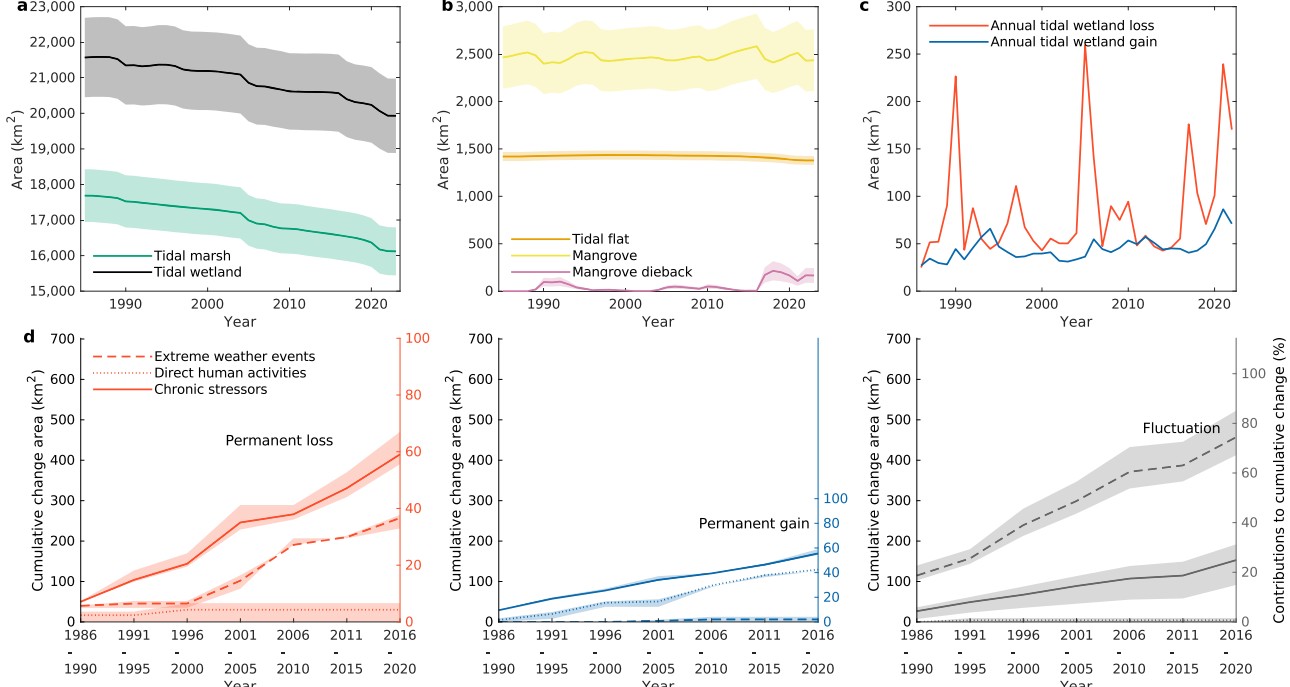

**Fig. 2 | Tracking tidal wetlands cover changes in the conterminous US (1985–2023). a** Annual area estimates of tidal wetland (including tidal marsh, mangrove, and tidal flat) and tidal marsh. **b** Annual area estimates of tidal flat, mangrove, and mangrove dieback. **c** Mapped annual tidal wetland loss and gain areas. **d** Major drivers (i.e., extreme weather events, direct human activities, and chronic stressors) contributions to permanent loss, permanent gain, and

fluctuation (including temporary loss, recovery, and internal transitions) of tidal wetlands. They share the same scale on the left y-axis to indicate the corresponding change area. The right y-axis indicates the proportion of the cumulative change associated with each driver. The x-axis indicates the five-year analysis period in which the change occurred. The shaded areas show 95% confidence intervals of estimates (**a–d**). Source data are provided as a Source data file.

## The limited but crucial role of direct human activities

Direct human activities were a minor driver of wetland loss, accounting for only 4% of the total observed losses (Fig. 2d and Supplementary Table S3). This confirms that since 1986, direct land conversion of tidal wetlands has been minimal in the conterminous US—a stark contrast to regions like Asia, where such transformations account for over two-thirds of losses[1]. This finding underscores the general effectiveness of US regulatory protections in preventing direct habitat destruction.

Conversely, direct human activities were the primary driver of tidal wetland gains, associated almost entirely with restoration projects. These projects comprised nearly 42% of all permanent gains observed in our study (Supplementary Table S3 and Supplementary Fig. 5b). However, our analysis reveals that these current, site-scale restoration efforts are insufficient to offset the extensive, accelerating losses driven by climate change. In Galveston Bay, for example, 4 km² of restoration has been dwarfed by 32 km² of loss over the past two decades[36]. In contrast, extensive restoration on the less-exposed Pacific coast, exemplified by the Napa River Salt Marsh Project (Fig. 1d), has led to net gains in the tidal marsh area. This success, however, occurs in a unique context: the region is generally free from hurricanes and has experienced the slowest rate of relative sea level rise in the

US[37,38], suggesting that restoration success is strongly contingent on the intensity of regional climate change pressures.

## Discussion

Our work provides a high-frequency view of a critical Earth system, revealing a continental-scale acceleration in the loss of US tidal wetlands. By producing the annual, 39-year map series, this study moves beyond static, epoch-based assessments to provide a continuous, dynamic view of coastal ecosystem change. The product can be useful for providing information for climate policy objectives, such as tracking tidal wetlands in national GHG inventory reports to the United Nations Framework Convention on Climate Change. Currently, in the US national GHG inventory, tidal wetland extent changes are estimated using the National Oceanic and Atmospheric Administration (NOAA) Coastal Change Analysis Program (C-CAP) maps that are updated every few years[39]. Analysis of six C-CAP maps from 1990 to 2015 also showed that with a high level of uncertainty[29], landward migration of coastal wetlands did not counter seaward losses[7]; the gross rate of annual loss at [−45.4 to −8.4] km² year⁻¹, was partially offset by gains of [10.6−21.5] km² year⁻¹[29]. Notably, C-CAP does not differentiate tidal coastal wetlands from nontidal coastal wetlands, introducing additional and

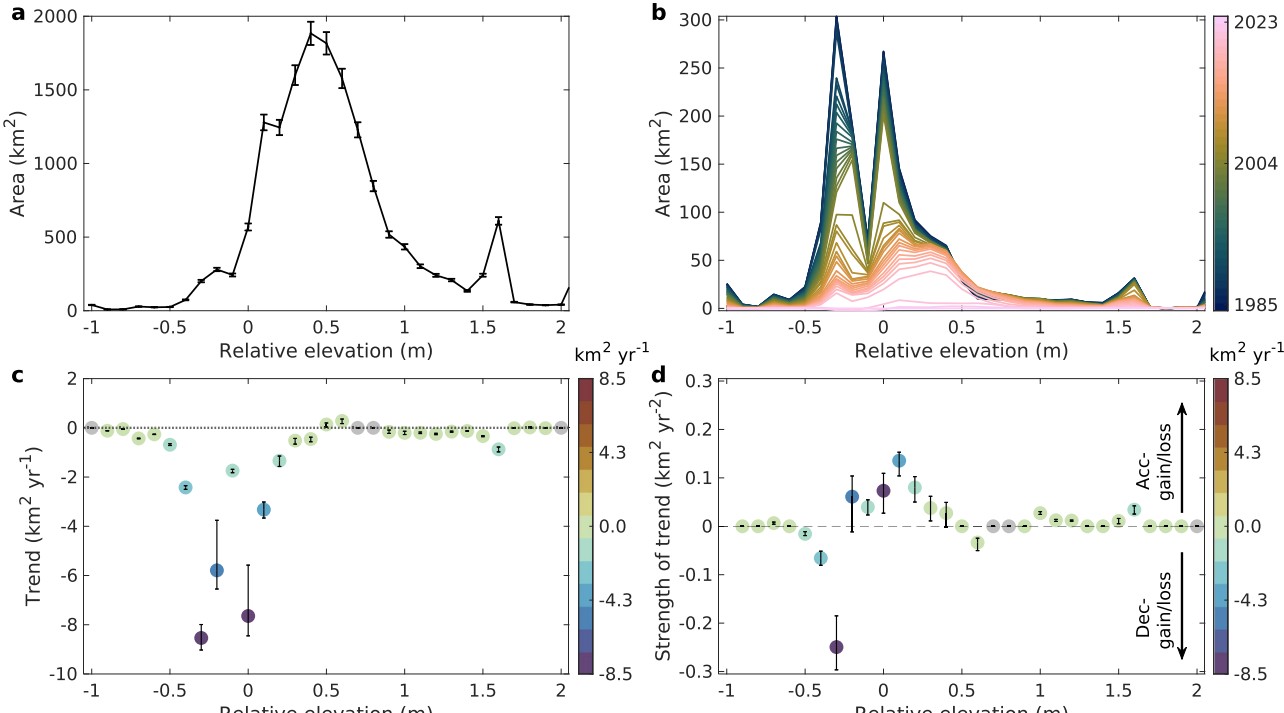

**Fig. 3 | Annual distribution of tidal marsh in the conterminous US, parsed by elevation. a** Area of tidal marsh in 2023 in 0.1 m elevation bins. **b** Difference in the yearly extent of tidal marshes relative to the 2023 area within each 0.1 elevation interval. The error bars show uncertainy at 95% confidence interval. **c** Annual gain or loss trend of the tidal marsh for every 0.1 m elevation range. **d** Acceleration and deceleration of the gain and loss trends for every 0.1 m elevation range in different coastal regions (refer to Supplementary Table S1). Positive trend strength indicates either decreasing losses or increasing gains of tidal marsh annually. Negative trend strength indicates either increasing losses or decreasing gain of tidal marsh annually. **c**, **d**, Temporal trends were estimated using Sen's slope and assessed for significance using the Mann-Kendall test at the 95% confidence interval. The elevation data are sourced from the USGS 3D Elevation Program (3DEP) Dataset, collected from around 2000 to 2015, and are referenced to the North American Vertical Datum of 1988 (NAVD88). That is, the elevation is a relative value based on a static state and not changed with sea level rise in the study period (refer to Supplementary Fig. S3). Y-axis scales vary by region. The statistics for different US coasts (Gulf of Mexico, Atlantic coast, and Pacific coast) are provided in Supplementary Fig. S2. Source data are provided as a Source data file.

substantial uncertainties into the analyses for soil carbon stock, carbon sink, and methane emissions[29]. Similarly, previous epoch-based approaches for the conterminous US during the period 2000–2020 have substantially underestimated the total cumulative tidal wetland loss (−232 km² versus −954 ± 36 km² in our map)[1]. Similarly, an epoch-based approach estimated tidal marsh loss of −561 km²[28], versus −942 ± 40 km² in our analysis. In contrast, the Global Mangrove Watch[11], which provides data from 1996 to 2020, overestimated mangrove loss in the US (−71 km² compared to our finding of −28 ± 4 km²). This discrepancy is primarily due to their omission of poleward expansion in Texas and Northern Florida. Additionally, it does not distinguish mangrove dieback from mangrove and therefore includes dieback areas within mapped mangrove extent. When mangrove dieback is explicitly accounted for, our analysis indicates a net expansion of mangrove-related extent (i.e., mangrove plus mangrove dieback) by 128 ± 70 km² from 1996 to 2020.

Another key finding in this study is that wetlands are not merely declining but that the rate of loss is accelerating, a critical and previously unquantified dynamic with profound implications for the long-term sustainability of coastal zones. This discovery is a direct result of our high-frequency monitoring capability, which allows the strength of trends (i.e., acceleration or deceleration) to be quantified at a continental scale. For example, at the regional scale, tidal marshes on the Atlantic coast exhibited relatively modest loss rates but a clear acceleration through time, underscoring the critical need to enhance ecological resilience and implement effective conservation strategies to mitigate further degradation[40,41]. This discovery provides a reality check on the effectiveness of current conservation strategies. Our results show that decades of successful regulatory protections, which

have been highly effective at preventing direct land conversion, are insufficient to protect tidal wetlands from the indirect and intensifying pressures of climate change[42]. The widespread nature of this accelerating loss indicates a systemic, national-scale vulnerability that has been largely invisible to lower-frequency monitoring approaches.

The most critical finding enabled by our tidal wetland cover change maps and sample analysis is the reversal in the dominant drivers of accelerating change. Our dense time series analysis, with a temporal resolution of approximately 32 days, captured the specific timing of cover changes and enabled the quantitative attribution of change drivers from map-stratified sample. While the chronic stressors, such as relative sea level rise, have caused the largest cumulative wetland loss, our maps show that the acute shocks of extreme weather events have emerged as the dominant force driving the acceleration of that loss. This does not suggest that these drivers act in isolation. Rather, our synoptic view provides crucial evidence that the interaction between them is changing. A plausible mechanism, supported by our data, is that long-term sea level rise renders marshes increasingly vulnerable (e.g., by reducing their elevation capital), while the increasing frequency and intensity of acute extreme weather events act as the trigger for catastrophic, large-scale loss events from which the ecosystem cannot recover.

This discovery about the drivers' underlying shifting dynamics in US tidal wetlands has profound implications for sustainability. Previous modeling-based approaches to predict tidal marsh resilience primarily considered chronic stressors of accelerating relative sea level rise and spatial variations in sediment supply. There is ongoing debate about whether the degree of vulnerability is under- or over-estimated for US tidal marshes based on modeling their adaptation potential in

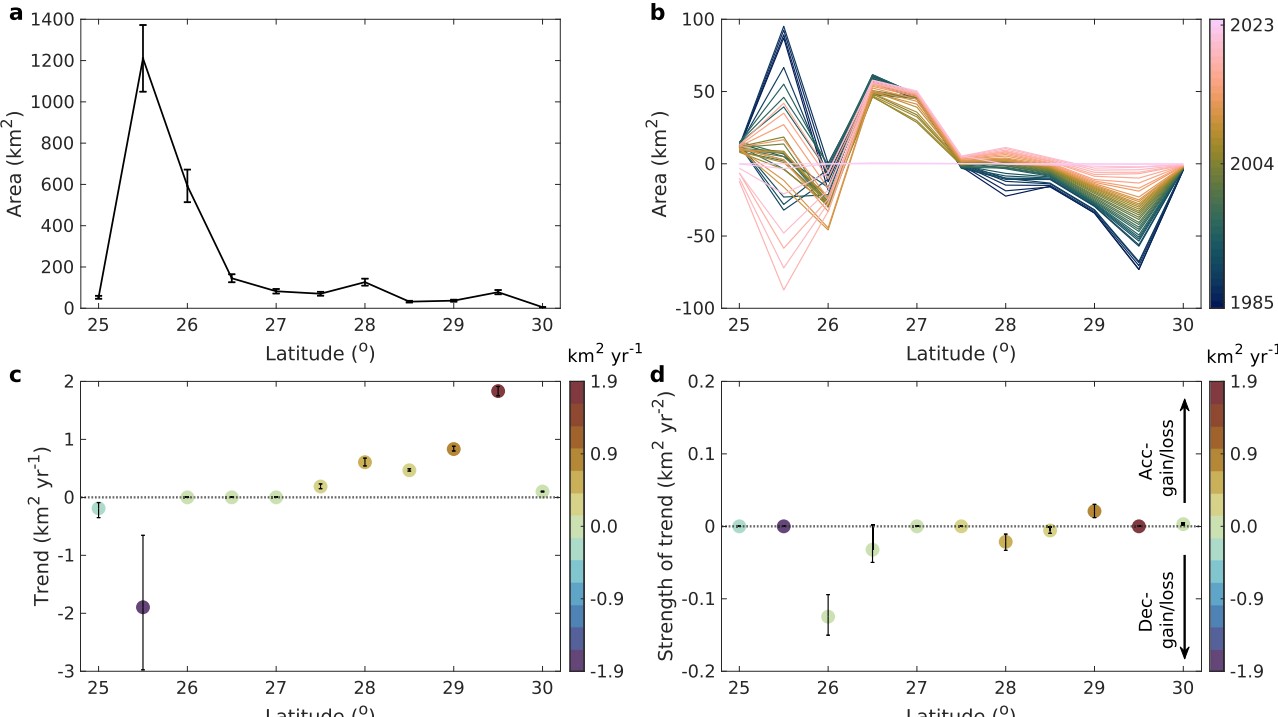

**Fig. 4 | Gain and loss of mangroves in the southern US. a** Area of mangrove in 2023 for every 0.5-degree of latitude. The error bars show uncertainty at 95% confidence interval. **b** Difference in the yearly extent of mangroves relative to area in 2023 within each 0.5-degree latitude interval from 1985 to 2023. **c** Annual gain or loss trend of mangroves for every 0.5-degree of latitude. **d** Acceleration and deceleration of the gain and loss trends for every 0.5-degree of latitude (refer to Supplementary Table S1). The positive strength of the trend indicates either fewer losses or more gains of mangroves annually, and the negative strength of the trend indicates either more losses or fewer gains of mangroves annually. **c, d** Temporal

trends were estimated using Sen's slope and assessed for significance using the Mann-Kendall test at the 95% confidence interval. Mangroves in the conterminous US located south of 26°N are experiencing a decreasing trend (such as the mangrove dieback in the Everglades depicted in Fig. 1h). By comparison, most northern regions (i.e., between 27°N and 30°N) are witnessing a significant acceleration in mangrove expansion (such as the mangrove encroachment into tidal marsh depicted in Fig. 1i). Acc- and Dec- indicate the acceleration and deceleration trend, respectively. Source data are provided as a Source data file.

horizontal (landward advance) and vertical (upward growth) directions[7,30,43–46]. The mid-Atlantic coast, with its significant relative sea level rise, frequent storm surges[47], and extensive marsh conservation efforts, exemplifies this debate, showcasing contrasting perspectives on marsh adaptation[41,48]. By directly analyzing historical satellite observations and their associated drivers, we move beyond theoretical modeling to provide empirical evidence of tidal marsh in response to compounded drivers of chronic stressors and acute shocks. Our findings demonstrate an alarming acceleration of tidal marsh loss during the past four decades, highlighting their increasing vulnerability at a national scale.

Compared with the widely investigated impacts from chronic stressors, the effects of extreme weather events on tidal wetlands are relatively underexplored. Existing research offers contrasting views on tidal wetland resilience[49,50] and vulnerability[8,51] to such events, often focusing on isolated, localized incidents[28,52]. Our findings highlight the critical need to consider the cumulative effects of repeated extreme events as well as the post-change recovery. Sustainability in the 21st century will require a paradigm shift towards managing resilience against acute shocks, such as extreme weather events. While storms had sometimes been viewed as beneficial to marshes through mineral sediment delivery and associated elevation gains[49,53,54], this assumption is increasingly challenged under conditions of rising storm frequency and intensity. Moreover, although total mangrove area across the conterminous US remained relatively stable between 1985 and 2023, this apparent stability masks growing vulnerability, as diebacks driven by hurricanes and severe winter freezes are mostly offset by poleward migration rather than true system resilience.

Reversing the accelerated loss trajectory of US tidal wetlands will require an increase in investment and a rethinking of coastal adaptation. Although regulatory protection has successfully limited direct conversion of tidal wetlands, these largely static approaches have proven insufficient to sustain tidal wetlands under accelerating sea level rise and increasing extreme weather, highlighting the need for adaptive restoration strategies[8,9]. First, the large areal historic conversions of tidal wetland since European colonization in the US[6], including an estimated ~4800 km² of restricted and impounded wetlands[55] and ~2400 km² of drained former wetlands[29], represent a major restoration opportunity, particularly where tidal connectivity can be re-established[32]. Multiple regional projects have actively restored tidal wetland ecosystems by removing and updating dike and impoundment systems to re-establish tidal exchange[56–59]. Restoration has also been achieved through managed wetland creation and sediment-based interventions implemented at regional scales[60]. Large-scale restoration programs in San Francisco Bay[61], for example, demonstrate how coordinated marsh creation, sediment enhancement, and phased tidal reintroduction can generate net tidal-wetland gains, illustrating the potential for proactive restoration to counterbalance climate-driven losses when applied at sufficient scale. In parallel, post-disturbance restoration following extreme events should be a critical component of adaptive management. Widespread mangrove dieback after major hurricanes[62,63], and storm-driven marsh degradation[64] highlight the importance of hydrological rehabilitation and assisted recovery to promote regeneration and reduce the risk of irreversible ecosystem transitions. Together, substantially increasing the extent, pace, and strategic targeting of restoration is essential to

reverse current loss trajectories and strengthen the long-term resilience of U.S. tidal wetlands, and our identification of loss hotspots offers a practical roadmap for prioritizing restoration actions.

While our large-scale analysis identifies the dominant proximate drivers of change (i.e., the immediate triggers of loss), disentangling the specific contributions of increased exposure (e.g., shifts in storm intensity regimes[37,65,66]) versus increased vulnerability (e.g., gradual loss of elevation capital[67–70]) remains a challenge. We hypothesize that the observed acceleration is likely the result of synergistic interactions between these factors: chronic stressors condition the landscape to be more vulnerable to acute shocks. However, separating their individual roles requires integrating complex meteorological data with localized marsh health metrics. Future work coupling this synoptic remote sensing record with process-based models has the potential to disentangle mechanisms[71]. For example, linking our observed hotspots of dieback and recovery with on-the-ground measurements of sediment accretion, porewater salinity, or nutrient loads can build multiple lines of evidence to explain the mechanisms of ecosystem collapse or survival observed from space. Importantly, the multi-decadal, observation-based record developed here can serve as an empirical constraint for process-based models that incorporate environmental and climatic variables, enabling more robust projections of future tidal-wetland trajectories under alternative climate and management scenarios[43,72]. Another important future direction is to move beyond binary cover conversion and explicitly quantify condition change and recovery capacity using dense satellite time series. Ecosystems often exhibit prolonged degradation and reduced recovery rates before crossing critical thresholds into irreversible states[73]. Time series indicators (such as time to recovery and disturbance severity) can provide early warning signals of approaching tipping points prior to permanent cover loss[34,74]. Integrating such condition-based metrics with change detection could therefore enable earlier identification of vulnerable tidal wetlands and inform proactive intervention before irreversible conversion occurs.

## Methods

### Mapped area

Our analyses were conducted along the entire coastal landscape in the conterminous US, covering an area of approximately 307,000 km$^2$. The coastal landscapes are defined using the widely used NOAA C-CAP maps[75] and elevation range (not greater than 10 m) derived from the US Geological Survey 3D Elevation Program (3DEP)[76]. To ensure all potential tidal wetland regions are included, we defined the study area with the maximum extent of palustrine wetlands, estuarine wetlands, open water and submerged lands with aquatic vegetation from the entire time-series of C-CAP maps. Tidal wetlands include tidal marshes, mangroves, and tidal flats[1]. Tidal marshes are defined according to the US National Wetlands Inventory (NWI)[77], which includes estuarine intertidal persistent emergent marshes that are either irregularly flooded or regularly flooded. It is important to note that tidal wetlands are characterized by heterogeneity, and the NWI assigns multiple codes for polygons containing marshes and other cover types. In this study, we regarded these polygons as tidal marshes if tidal marshes were part of the ecosystem. Tidal flats are a major intertidal ecosystem that interacts closely with mangroves and tidal marshes in terms of hydrology, sediment exchange, and ecosystem processes[58,78]. As such, we include tidal flats within the tidal wetland framework[1,79] adopted in this study.

### Input data

To study the extent and changes in tidal wetlands across the conterminous US, we used a total of 176,233 Landsat images and many ancillary data. We downloaded all available Landsat 4–8 Collection 2 Analysis Ready Data (ARD)[80] from the US Geological Survey for the years between January 1984 and April 2024 in the conterminous US,

with cloud cover less than or equal to 80%[81]. We excluded Landsat 1–3 (MSS) data because differences in data resolution, temporal density, and radiometric consistency relative to later sensors limit their suitability for long-term time-series analysis. The main inputs for our analysis were the surface reflectance of six spectral bands, including blue, green, red, near infrared (NIR), and two shortwave infrared (SWIR) bands, along with the thermal and quality assessment bands[82]. Landsat observation frequency varied through time due to single- and bi-satellite acquisition periods and spatially due to path overlap. We standardized observation frequency to a near-monthly level. All clear observations were used during single-satellite periods, whereas during bi-satellite periods observations were composited every 32 days[81]. When clear scenes exceeded this density, a maximum of 12 observations per year was randomly selected. Composites were generated using the maximum NIR-to-blue ratio[83].

Tides stand as a primary driver for short-term fluctuations in water levels, exhibiting distinct variations based on specific times and locations throughout the year. To estimate the tidal fluctuations at the time of Landsat image acquisition, we incorporated the water levels provided by a global ocean tide model, namely the latest empirical ocean tide (EOT20) model[84]. EOT20 provides ocean tide predictions by using the land-sea mask from the generic mapping tools (GMT) data. We extracted the water level height prediction values for each pixel at the time of each Landsat image acquisition. Some upland regions may not be covered by the tide prediction model. In these pixels, we employed the tidal conditions of the nearest pixel with valid predictions[18].

We used the high spatial resolution elevation data (10 m) derived from the USGS 3D Elevation Program (3DEP) for our analyses. The 3DEP Digital Elevation Model (DEM) product has an absolute vertical accuracy of approximately 0.82 meters root mean square error (RMSE) across the US, which has a higher accuracy than that of widely used global DEM products, such as NASADEM (3.30 m) and Shuttle Radar Topography Mission (3.79 m)[85] at a spatial resolution of 30 m. This elevation information is pivotal in assessing the distributions of tidal marshes along the gradient from seaward to landward direction.

### Dense-time-series-based mapping

We used dense time series model, namely detection and characterization of coastal tidal wetlands change (DECODE)[18], to address the confounding effects of tidal fluctuations in the remote sensing of tidal wetlands. DECODE is a per-pixel-based approach and monitors the trajectory of pixels over time, allowing for the detection of spectral breaks at each observation time (refer to Supplementary Figs. S4–6 for examples). For each pixel, DECODE detects spectral breaks (i.e., an abrupt change) when a sequence of multiple consecutive Landsat observations significantly deviates from their predicted values[86]. This enables improvements compared to datasets derived from traditional approaches[1,7,8,12,17,87], including (i) the direct change detection that avoids compounded errors by differentiating classification maps at different times, and (ii) identification of the timing of the change at a temporal resolution of Landsat satellite revisit cycle[88,89].

We used spectral-temporal features derived from the time series model as variables to conduct the cover classification, which avoided mapping uncertainty caused by tidal fluctuations[90–92]. Coefficients and RMSE, from all seven Landsat bands (including the thermal band brightness temperature) and five wetland-related indices, serve as input variables for constructing a random forest classifier. The spectral breaks divided the time series into different temporal segments. Each temporal segment indicates either a stable land cover state or gradual changes of the cover conditions (e.g., recovery or decline of the vegetation). We followed the procedure of "segment cover type recognition"[18] to generate either the "one class per segment" or "transitional covers" for each temporal segment. "One class per segment" designates a stable state for the segment with a

unique land cover. "Transitional covers" reflect a gradual and irreversible change in the temporal segment, such as the conversion from tidal marsh to open water due to its drowning with relative sea level rise, and the gradual encroachment of mangrove to tidal marsh. We collected a training dataset containing 29,487 Landsat pixels derived from visual interpretation of very-high-resolution imagery (Google Earth and PlanetScope), Landsat time series, and auxiliary datasets (C-CAP and NWI). Six classes were defined: tidal marsh, mangrove, tidal flat, mangrove dieback, open water, and other coastal land cover. The "other" class (e.g., upland vegetation, cropland, developed/barren land, mixed shoreline pixels) was first classified as separate subclasses and merged in the final maps. Mangrove dieback was treated as a distinct class and defined as severely degraded mangrove persisting for more than one growing season, distinguishable spectrally and temporally from healthy mangrove cover[34].

All the annual maps are post-processed with "object-based rules"[16,93]. A minimum mapping unit of four 30 m × 30 m Landsat pixels[94,95] is used to filter the "salt and pepper" noise in the pixel-based analysis. Additionally, a contextual algorithm, based on the spatial relationship between tidal wetlands and coastal open water, is used to remove the misclassified tidal wetlands. The predicted tidal wetland objects, far away from open water, are removed from annual cover maps. The distance threshold is determined by the size of the tidal wetland objects, given that a larger patch corresponds to a larger distance threshold, considering that larger patches have lower misclassification likelihood.

## Temporal trend analyses

We employed a combination of the Sen's slope estimator and the Mann–Kendall (MK) statistical test (95% CI)[96] on annual tidal wetland extent maps to identify and quantify linear trends of tidal wetland area across different spatial scales. To evaluate the spatial patterns of cover change trends, we compiled Landsat pixels into subbasin hydrological units, derived from the USGS Watershed Boundary Dataset, with a total number of 213 units that contain tidal wetlands across the conterminous US. Monitoring the poleward migration of mangroves and the landward migration of tidal marshes is crucial for understanding the impacts of climate change and how tidal wetlands adapt and respond. Mangrove migration was tracked along latitudes (25°N–31°N) with 0.5° intervals, while tidal marsh migration was assessed along elevation (−1 to 2 meters) with 0.1 m bins.

Using the Sen's slope estimator and MK test, we quantified the linear trend in annual tidal wetland area (Supplementary Fig. S9a). To measure the acceleration/deceleration of tidal wetland gain and loss, we used a 10-year moving window-based approach[97,98], in which 30 epochs from 1985–1994 to 2014–2023 were used in this analysis. We calculated the trends by using the Sen's slope estimator for each 10-year window (Supplementary Fig. S9b). Subsequently, a second level of the Sen's slope estimator and MK test was used to represent the acceleration/deceleration based on the changes in trends. The trends of gain/loss, along with the acceleration and deceleration rates of the target cover type within the spatial patch, are illustrated in Supplementary Table S1.

## Driver analysis of tidal wetland cover change

We quantified the contribution of each driver to cumulative tidal wetland cover change and its acceleration or deceleration (Fig. 2d and Supplementary Table S3). The DECODE approach separates abrupt changes (acute shocks, such as extreme weather events or human activities) via spectral breaks and gradual changes (chronic stressors, such as sea level rise, sediment supply shifts, nitrogen loading, and pollution) via regression slopes. Yet these distinctions are not absolute: rapid sea level rise can manifest as abrupt shifts when the magnitude of slope changes, while severe nitrogen loading and pollution may also appear as abrupt spectral breaks (refer to an example in

Supplementary Fig. S4a). Therefore, by leveraging the temporal details of our tidal wetland map time series, we used the sample-based approach to interpret the proximate drivers (i.e., human activity, extreme weather events, chronic stressors) and quantify their impacts on tidal wetlands. We generated random sample from the change areas every five years, spanning from 1986 to 2020. These sample were then manually interpreted to assess their accuracy and identify the underlying drivers.

We investigated three types of tidal wetland cover changes: permanent gain (Supplementary Fig. S5), permanent loss (Supplementary Fig. S4), and fluctuations (Supplementary Fig. S6). Permanent gain refers to areas where tidal wetlands have been established in new places, while permanent loss indicates areas where tidal wetlands have been lost and not recovered over the past four decades. Fluctuations include temporary changes, such as post-disturbance recovery, and internal transitions like mangrove encroachment of tidal marsh. To determine the contribution of different drivers on these three categories of wetland cover changes, we randomly generated multi-phase stratified sample with three strata: permanent loss, permanent gain, and fluctuations. We generated seven sets of stratified samples every 5 years from 1986 to 2020, with 20 random sample for each change type per period, weighted by the amount of annual cover change area during each 5-year period. For each period, the number of changes attributed to each driver, combined with their weights, determined the proportion of cover change areas related to each driver. We then calculated the total cumulative proportion from 1986 to 2020 for each driver, indicating the percentage of the total cumulative tidal wetland cover change areas associated with each driver (refer to Supplementary Table S2). Additionally, our multi-phase estimates revealed the acceleration and deceleration of tidal wetland cover changes attributable to each driver. We calculated the strength of the trend in cumulative change proportion to indicate their positive or negative contributions to the acceleration/deceleration of tidal wetland changes (Supplementary Table S2). We then manually interpreted drivers of these cover change sample with reference to multiple data sources. Further, 1000 iterations of bootstrapping are used for uncertainty estimates (95% confidence interval) of the driver analyses[99].

Using historical Landsat observations (1982–2024), we identified long-term trends and the timing of changes, as well as understanding how the cover change occurred (i.e., chronic or abrupt). For example, we can interpret whether an extreme weather event occurred before the change point from historical records, such as climate and weather databases (e.g., NOAA hurricane tracks database and drought events database), and local ecosystem restoration documentation. High-resolution historic images from Google Earth also aided our interpretations. Notably, the beginning and end years of the sample-based analysis are 1986 and 2020, respectively. The Landsat observations collected from 2021 to 2024 are used to verify the sample from the map-derived permanent loss stratum, which could be biased in the last several years when they are still in the recovery stage. For instance, tidal wetlands damaged by Hurricane Irma in 2017 and Hurricane Laura in 2020 are still recovering. While current maps might classify these areas as permanent loss, based on the interpretation of the additional four years of Landsat observations, we can correctly label them as fluctuations due to ongoing recovery (refer to an example in Fig. S8). Similarly, we can use Landsat observations collected from 1982 to 1985 to confirm that samples from the map-derived permanent gain stratum represent newly established tidal wetlands rather than temporary recovery changes.

## Accuracy assessment and bias adjustment

All land cover land change maps developed from satellite images have errors and biases. We followed the "good practice" recommendations in land change analyses and applied an unbiased estimator to adjust area estimates and provide the confidence intervals[100,101]. An

independent set of land cover reference sample (2028 pixels) was generated across the study area based on the stratified random sampling strategy with the annual tidal wetland cover maps from 1985 to 2023 as the stratum. In this stratification, the individual reference sample represents not only a location on the ground but also a place in time. The number of sample for each category is calculated based on their area proportion in the final map, with a minimum number of 150. A confusion matrix (Supplementary Table S4) was firstly computed based on the reference sample and map product and then translated into terms of unbiased accuracies and area proportions using the post-stratified estimator presented in the "good practice" recommendations[101]. The overall accuracy of the tidal wetland maps is $99.46 \pm 0.26\%$. Producer's accuracies (reflecting omission error) for tidal marsh, mangrove, and tidal flat are $97.59 \pm 2.67\%$, $91.91 \pm 12.23\%$, and $100.00 \pm 0.00\%$, respectively, while user's accuracies (reflecting commission error) are $94.59 \pm 3.07\%$, $99.34 \pm 1.09\%$, and $94.08 \pm 3.16\%$, respectively. These results indicate that tidal marshes and tidal flats are slightly overestimated in the maps, with error-adjusted areas equal to $96.93 \pm 4.06\%$ and $94.08 \pm 3.16\%$ of their mapped areas. In contrast, mangroves are slightly underestimated, with an error-adjusted area of $108.08 \pm 14.42\%$ of the mapped value.

We also evaluated the accuracy of the tidal wetland cover changes using the multi-phase, stratified random sample generated for the driver analyses. Four cover change strata were defined, including three tidal-wetland-related change types (i.e., permanent gain, permanent loss, and fluctuations) and an "others" category. For each phase, 20 random samples were allocated to each change-related stratum and 60 sample to the "others" stratum, yielding 120 samples per phase and 840 samples in total. Sample allocation among strata was weighted by the mapped change area within each five-year period to ensure adequate representation of rare but ecologically important change processes. A confusion matrix (Supplementary Table S5) was first constructed using the reference sample and map products and then translated into unbiased accuracy and area estimates using the post-stratified estimator recommended by established "good practice" guidelines[101]. The producer's accuracies (reflecting omission error) for permanent gain, permanent loss, and fluctuations are $99.00 \pm 1.19\%$, $96.21 \pm 4.21\%$, and $91.21 \pm 7.51\%$, respectively, while the user's accuracies (reflecting commission error) are $61.03 \pm 8.33\%$, $59.21 \pm 8.27\%$, and $75.76 \pm 7.45\%$, respectively. These results indicate that cover-change classes are generally overestimated in the maps, with error-adjusted areas equal to $61.6 \pm 8.4\%$, $61.5 \pm 8.8\%$, and $83.1 \pm 10.1\%$ of the mapped change areas for permanent gain, permanent loss, and fluctuations, respectively. Notably, for rare land cover change classes, omission error is generally considered more consequential than commission error, as missed change cannot be recovered in subsequent analyses, while commission bias can be mitigated through post-processing or design-based area correction[101,102].

The reference sample was interpreted blind to map labels to avoid confirmation bias, on the reference of high-resolution historical images in Google Earth and time-series Landsat observations (1985–2024). Notably, mixed pixels are common in tidal wetland environments given the 30 m spatial resolution of Landsat imagery. During validation, pixels located along class boundaries were interpreted using a fractional-cover criterion. A pixel was labeled as tidal wetland when tidal-wetland cover occupied approximately ≥20% of the pixel area and was visually identifiable as the dominant ecological feature. Validation points falling along boundaries between two tidal-wetland classes that could not be confidently assigned were excluded and replaced with other randomly selected locations to avoid ambiguous class attribution.

## Reporting summary
Further information on research design is available in the Nature Portfolio Reporting Summary linked to this article.

## Data availability
All data used in this study were obtained from open data sources. The Landsat Collection 2 US Analysis Ready Data (ARD) data were downloaded from USGS https://earthexplorer.usgs.gov/. Tide predictions were obtained from global ocean tide models (EOT20) developed at DGFI-TUM https://www.seanoe.org/data/00683/79489/. Historical maps of the NOAA C-CAP Regional Land Cover and Change maps were obtained from https://coast.noaa.gov/digitalcoast/data/ccapregional.html. The seamless 3DEP DEM dataset for the US were provided by USGS and downloaded from Google Earth Engine (https://developers.google.com/earth-engine/datasets/catalog/USGS_3DEP_10m). The National Wetlands Inventory (NWI) dataset (https://www.fws.gov/program/national-wetlands-inventory) was employed to help interpret the training and validation data. US boundary data were from the US Census Bureau (https://www.census.gov/geographies/mapping-files/time-series/geo/carto-boundary-file.html). To interpret the drivers associated with the extreme weather events, historical hurricane tracks (https://coast.noaa.gov/hurricanes/) and National Integrated Drought Information System (NIDIS) (https://www.drought.gov/) in NOAA platforms are used. Tidal wetland annual extent, loss and gain, and change derived from our DECODE approach and supporting our analyses are available at https://gers.users.earthengine.app/view/tidalwetlandcover.

## Code availability
The custom MATLAB code used for the approach, data processing and analyses is available via GitHub at https://github.com/GERSL/DECODE.

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

## Acknowledgements

This study was supported by USGS North Atlantic coast Cooperative Ecosystem Studies Unit (CESU) Program for "Detection and Characterization of Coastal Tidal Wetland Change (G19AC00354)" and "A Sample-based Approach for Analyzing the Driver of Coastal Tidal Wetland Changes (G23AC00410)", US Fish and Wildlife Service Program for "Evaluating Coastal Tidal Wetland Change in the Conterminous United States (F22AP02176)", USGS Coastal and Marine Hazards and Resources Program, and NASA Carbon Monitoring System Program for "An Alkalinity and Inorganic Blue Carbon Monitoring System: Crediting Wetland-to-Ocean Lateral Fluxes in Carbon Markets". Any use of trade, firm, or product names is for descriptive purposes only and does not imply endorsement by the US Government. The findings and conclusions in this article are those of the authors and do not necessarily represent the views of the U.S. Fish and Wildlife Service.

## Author contributions

X.Y.: conceptualization, methodology, software, validation, analysis, investigation, data curation, writing-original draft, writing-review and editing, visualization. S.Q.: software, analysis, investigation, data curation, writing-review and editing. K.K.: conceptualization, validation, resources, analysis, writing-review and editing. Z.L.Z: conceptualization, validation, resources, writing-review and editing. S.C.: resources, writing-review and editing. N.M.: analysis, writing-review and editing. Z.Z.: conceptualization, methodology, software, analysis, resources, writing-original draft, writing-review and editing, supervision, project administration, funding acquisition.

## Competing interests

The authors declare no competing interests.

## Additional information

**Supplementary information** The online version contains Supplementary material available at https://doi.org/10.1038/s41467-026-71464-2.

