## [Transparent Peer Review File · Nature Communications]

The accelerating loss and shifting dynamics of US tidal wetlands

Corresponding Author: Dr Zhe Zhu

Version 0:

Reviewer comments:

Reviewer #1

(Remarks to the Author)

The study presents an expansion of the group's previous work on Florida and Northeast, utilizing time series analysis and USGS ARD data for Landsat 5, 7, and 8. The approach uses the DECODE algorithm, which incorporates tidal information into the time series analysis, along with random forest classification of wetlands and an analysis of drivers. The manuscript details a robust remote sensing analysis of tidal wetlands and tidal flats. The study would benefit from revisions focused on direct and clear language highlighting the many important and interesting results but reducing sensational language, e.g., "motion picture" to refer to the time series or "the vital fingerprint of climate change" to refer to abrupt or gradual time series changes. Additional citations for context are necessary especially in the discussion. This is a good manuscript that, with adjustments to the accuracy assessment and improvements to the writing, will make a great contribution to the literature. Tidal flats should not be included as coastal wetlands or there should be a justification for their inclusion.

Mangrove analysis largely overlaps author's previous published work and that study should be cited and then the current papers discussion should highlight new findings from the evaluation of tidal wetlands together.

The authors should clarify how their independent set of reference points were evaluated and classes determined. The overall accuracy of 99.42% is high and ultimately rather uninformative. While coastal ecosystems are defined it is unclear how a 30 m pixel is assessed, i.e., how were individual pixels evaluated what % of an ecosystem was necessary to evaluate it as that ecosystem. Tidal marsh having the fewest reference points does not make sense giving the goals of the research and what is trying to be conveyed by the accuracy assessment. As currently distributed, the overall accuracy conveys the ability of the classifier to successfully identify other and water classes. Ideally, the accuracy assessment would include change categories as this manuscript is a change analysis.

The discussion is too focused on the importance of this work and would be strengthened with additional citations surrounding the current policies, proposed policies, modeling, and in situ research that also supports many of the statements in the discussion.

Line 29 – Ecosystem services can be illustrative but comparisons such as these seem reductive and not particularly illustrative of a particular point i.e., instead of using ecosystems services discuss in more detail why tidal wetlands are more valuable to humans than tropical forests?

Line 33 – Peculiar wording. Land use change "changes in land use, resulting in widespread wetland filling." Wetland filling is the process through which the land use change is happening.

Line 43 – are they truly protected? Protected area vs. the minimal and eroding protections provided by Clean Water Act should be clarified. The protection of these ecosystems is brought up again in the discussion and exactly what is protected and how should be clear from the first mention.

Line 60 – Landsat 1 launched in 1972, while there are obvious reasons the analysis does not and is not expected to include the MSS era Landsat data they are nevertheless a part of the Landsat archive and should not be erased by the sentence phrasing.

Line 65 – "...revealing a reality that is far more alarming than previously understood." seems a little out of touch with the literature both in situ and local to regional remote sensing studies including the author's own work (Potter 2021; Watson et al. 2017; Vinent et al. 2021; Yang et al. 2024; Di Vittorio et al 2025; Ganju et al. 2025). The writing is suggesting a shocking new finding but in reality, this is additional evidence supporting the previous observations and unifying them behind a robust remote sensing framework.

Line 93 – "... the vital fingerprint of climate change..." unclear at this point in the text how Figure 1. Illustrates the vital fingerprint of climate change or what exactly is being referred to is it temperature, sea level rise, droughts, hurricanes?

Line 107 – "...Coast of Gulf marsh loss" change to "Gulf coast marsh loss" or some other term.

Line 112 – “...Louisiana estuary...” Unclear what is being referred to here as this is not the name of an estuary as far as I am aware. A particular estuary should be referred to by name.

Line 169 – Greater detail on this impediment to migration including how forest resilience to SLR might be a factor.

Line 270 – “Our findings suggest that a conservation paradigm focused on static protection and mitigating chronic drivers is no longer sufficient.” without citation it is very unclear who or what the authors are referring to. The authors should identify exactly what policies and practices their research identifies as being inadequate and where.

Line 280 – The sentence seems to conflate the time series derived abrupt vs. gradual which are spatial and the interpreted classes which are referred to as drivers but not spatially derived. Partially clarified in the subsequent paragraph, it still needs to be clearer in the text.

Line 445 – remove “other” as 3DEP is not a global DEM. A better comparison would be DeltaDTM or other global coastal DTM products (Pronk et al. 2024).

Figure 1. The trends texture visualization makes the figure hard to interpret quickly and difficult to see differences in clusters of small estuaries.

Potter, C., 2021. Remote sensing of wetland area loss and gain in the western Barataria Basin (Louisiana, USA) since Hurricane Katrina. *Journal of Coastal Research*, 37(5), pp.953-963.

Di Vittorio, C.A., Wiles, M., Rabby, Y.W., Movahedi, S., Louie, J., Hezrony, L., Cifuentes, E.C., Hinchman, W. and Schluter, A., 2025. Mapping coastal wetland changes from 1985 to 2022 in the US Atlantic and Gulf Coasts using Landsat time series and national wetland inventories. *Remote Sensing Applications: Society and Environment*, 37, p.101392.

Ganju, N.K., Ackerman, K.V., Defne, Z., Mariotti, G., Curson, D., Posnik, Z., Carr, J.A. and Grand, J., 2025. A Simple Predictive Model for Salt Marsh Internal Deterioration Under Sea-Level Rise and Sediment Deficits: Application to Chesapeake Bay. *Estuaries and Coasts*, 48(6), p.178.

Vinent, O.D., Herbert, E.R., Coleman, D.J., Himmelstein, J.D. and Kirwan, M.L., 2021. Onset of runaway fragmentation of salt marshes. *One Earth*, 4(4), pp.506-516.

Yang, X., Zhu, Z., Kroeger, K.D., Qiu, S., Covington, S., Conrad, J.R. and Zhu, Z., 2024. Tracking mangrove condition changes using dense Landsat time series. *Remote Sensing of Environment*, 315, p.114461.

Pronk, M., Hooijer, A., Eilander, D., Haag, A., de Jong, T., Voudoukas, M., Vernimmen, R., Ledoux, H. and Eleveld, M., 2024. DeltaDTM: A global coastal digital terrain model. *Scientific Data*, 11(1), p.273.

(Remarks on code availability)

Previously I reviewed the code. It worked as expected and is well documented.

Reviewer #2

(Remarks to the Author)

This is a welcome and important contribution and I unreservedly recommend publication. The potential impact of sea-level rise on coastal wetlands has been a major theme of research in recent years, and good progress has been made understanding the tipping points associated with mangrove and tidal marsh survival. This paper provides a comprehensive analysis of tidal marsh and mangrove habitat change at high temporal and spatial resolution for the conterminous USA. The United States is a fascinating case study, given strong regional differences in the rate of sea-level rise and exposure to extreme storms. The results provide perhaps the first detailed continental-scale assessment of the relative contributions of sea-level rise, extreme storms and anthropogenic stressors in shaping the extent of these important wetlands. The finding that sea-level rise is historically the dominant driver, but that extreme storms are increasing in their relative contribution is a very important finding. As the authors argue in their concluding statements, the two drivers are increasingly interacting in driving a trend of reduced tidal marsh extent. Mangrove low latitude losses appear offset by poleward expansion for now. These findings align with projections of theoretical studies, but it is very exciting (if disturbing) to see clear data demonstrating these patterns.

One suggestion that might not be too onerous. The findings give rise to the question of whether the increased losses to storms are due to increased exposure (i.e. an increase in the frequency and intensity of storms) or increased vulnerability (lower tidal frame due to sea-level rise), or both. One way of considering this is to look at data relating to storm frequency/intensity trends in these regions. Might this be considered?

Secondly, I was a little surprised to find that several recently published observational and theoretical papers published in leading journals (e.g. papers in *Science*, and *Nature* since 2020) providing context for these observations were not cited. A quick Google Scholar search will identify these. Some of the referencing seems a little dated.

Beyond these two suggestions, my observations are of a minor nature, as follows:

Line 58: “Here, we provide the definitive, observation-based answer to this question”. Make it clearer what this question is.

Line 80 (figure caption). There is no such place as “Louisiana estuary”. Perhaps “estuaries of Louisiana?”

Line 81. Instead of “seaward erosion” I would suggest “seaward edge erosion”. The former gives the sense that the marsh is eroding in a seaward direction.

Line 82. It would be good to give some coordinates for this figure

Line 89. I suggest you cite the loss here as negative (-19,931) in common with the other estimates

Line 93. ...year-2. Shouldn't this just be year-1?

Line 124. There is an apparent contradiction in the wording here: "...show only marginal declines in loss rates but are marked by accelerating trajectories of loss". How is the rate different to the trajectory?

211. You don't mention extreme freezing events as a driver of mangrove dynamics in the Gulf. This seems odd given the voluminous literature on this issue.

Line 273. "The notion that storms can be beneficial to marshes must be re-evaluated...". This seems to need a reference to a paper making this claim.

Line 278. You mention the importance of identifying and supporting corridors for landward expansion here, a point often made. However, do your results support the contention that these corridors are actually being used in any widespread way?

Line 589. The photos in (b) are a little too grainy in my version

Line 590. Spelling "Hurricane Katrina that led to.." not "lead"

(Remarks on code availability)

Reviewer #3

(Remarks to the Author)

It was a pleasure to review the manuscript by Yang et al., which provides an insight into the change in saltmarsh extent in the USA. I found the manuscript to be interesting and generally well-written. However, the manuscript requires extensive revisions before it can be considered for publication.

Currently, the manuscript has too much focus on the USA. As a result, it is at risk of being too parochial and lacking wider appeal. For example, I would recommend the authors incorporate further international examples into the introduction to increase the global application and relevance of the work.

The novelty of study has been other emphasised. We know that marshes are not stable and other studies have investigated the change in extent of saltmarshes both in the USA and elsewhere. The novelty comes from how the authors have considered tidal variability and it is important that the authors do not "overplay" the originality of the work.

The discussion requires considerable editing, especially as it currently lacks wider context. This mainly comes from the lack of citations in the discussion, but further consideration of the wider application and implications of the work would also be beneficial.

(Remarks on code availability)

Version 1:

Reviewer comments:

Reviewer #1

(Remarks to the Author)

The authors have addressed my previous comments comprehensively. I recommend the manuscript for publication.

(Remarks on code availability)

I previously reviewed the code.

Reviewer #2

(Remarks to the Author)

The authors have undertaken a comprehensive review of the manuscript in my view adequately addressing the issues raised by each of the reviewers. As suggested in my original review, the work is novel and important, and deserving of publication now that these concerns have been addressed. Could the authors please check line 64- should this read Landsat or Landsats? (the singular?)

(Remarks on code availability)

Reviewer #3

(Remarks to the Author)

Following the revisions made after the first review, the manuscript by Yang et al is much improved. It is my view that the manuscript should be considered for publication, but only after the reviews re-visit the global context of their work. I acknowledge the amends made by the authors following my comment in the first review regarding the focus on US tidal wetlands but think they could go further contextualise and quantify global loss. For example, recent reports such as Brook et al. (2025) have attempted to quantify global saltmarsh loss (<https://www.wwf.org.uk/sites/default/files/2025-06/state-of-the-worlds-saltmarshes-2025.pdf>). I would like to see the authors incorporate some of these estimates into their introduction to

ensure there is global context. The same is true for the abstract, which provides little awareness of the 'bigger picture' ie tidal wetland loss on a global scale.

(Remarks on code availability)

Reviewer #1 (Remarks to the Author):

Comment 1.1 The study presents an expansion of the group's previous work on Florida and Northeast, utilizing time series analysis and USGS ARD data for Landsat 5, 7, and 8. The approach uses the DECODE algorithm, which incorporates tidal information into the time series analysis, along with random forest classification of wetlands and an analysis of drivers. The manuscript details a robust remote sensing analysis of tidal wetlands and tidal flats. The study would benefit from revisions focused on direct and clear language highlighting the many important and interesting results but reducing sensational language, e.g., "motion picture" to refer to the time series or "the vital fingerprint of climate change" to refer to abrupt or gradual time series changes. Additional citations for context are necessary especially in the discussion. This is a good manuscript that, with adjustments to the accuracy assessment and improvements to the writing, will make a great contribution to the literature.

Response: We sincerely appreciate these constructive suggestions, which have helped improve the clarity and rigor of the manuscript. In the revised version, we have expanded the Discussion with additional citations as suggested to better situate our findings within the broader tidal-wetland literature. The discussion now includes relating our advances in US tidal wetland monitoring to link to prior process-based analyses of tidal-wetland resilience and enhance our discussion of the implications of synoptic monitoring for coastal restoration and management.

We have also clarified and strengthened the accuracy assessment by explicitly describing our validation framework following established “good practice” guidelines (Olofsson et al., 2014), and by conducting a dedicated accuracy assessment for the tidal-wetland cover change.

Lastly, as requested we have revised the vague descriptions based on this suggestion and edited the full manuscript to avoid the use of sensational language.

Comment 1.2 Tidal flats should not be included as coastal wetlands or there should be a justification for their inclusion.

Response: Thank you for this thoughtful comment. Whether tidal flats should be included within coastal (tidal) wetlands depends largely on the definition adopted. While some studies restrict tidal wetlands to vegetated intertidal ecosystems (e.g., mangroves and tidal marshes), others also include non-vegetated intertidal systems such as tidal flats. In this study, we include tidal flats within the tidal wetland framework because they constitute a major intertidal ecosystem and are tightly coupled with mangroves and tidal marshes through shared hydrology, sediment exchange, and biogeochemical and ecological processes. This definition is consistent with those adopted by Murray et al. (2022, Science) and has also been used in subsequent analyses (Saintilan et al., 2023, Nature) and research perspectives (Campbell et al., 2022, Nature). In addition, some studies in remote sensing (Zhang et al., 2022a), geophysics (Zhang et al., 2022b), ecology (He et al., 2025a), as well as a recently published review paper (He et al., 2025b) likewise treat tidal flats as an integral component of tidal wetlands.

We have clarified the coastal wetland definition in the manuscript and cited the papers above to justify their inclusion (Lines 380-383): **Tidal flats are a major intertidal ecosystem that interacts closely with**

mangroves and tidal marshes in terms of hydrology, sediment exchange, and ecosystem processes^{76,77}. As such, we include tidal flats within the tidal wetland framework^{1,78} adopted in this study.

Comment 1.3 Mangrove analysis largely overlaps author’s previous published work and that study should be cited and then the current papers discussion should highlight new findings from the evaluation of tidal wetlands together.

Response: Done. We have cited this paper (Yang et al 2024) in the revision as suggested and advanced our discussion. The Yang et al (2024) paper focused on Florida mangroves subjected to repeated hurricane disturbances and introduced an algorithmic extension, namely DECODER (DECODE and Recovery), designed specifically to characterize mangrove disturbance, recovery, and resilience (mangrove conditions). As such, the mangrove study primarily represents an algorithm development and resilience analysis in Florida.

In contrast, the present manuscript applies the DECODE framework to produce a continental-scale assessment of tidal wetland cover change across the conterminous US. While our previous work focused on condition change (recovery/resilience) in Florida, this study focuses on the timing of permanent cover conversion and the attribution of dominant drivers at a national scale. We now clearly distinguish these scopes in the revised Discussion (**lines 358-365**): **Another important future direction is to move beyond binary cover conversion and explicitly quantify condition change and recovery capacity using dense satellite time series. Ecosystems often exhibit prolonged degradation and reduced recovery rates before crossing critical thresholds into irreversible states⁷¹. Time series indicators (such as time to recovery and disturbance severity) can provide early-warning signals of approaching tipping points prior to permanent cover loss^{32,72}. Integrating such condition-based metrics with change detection could therefore enable earlier identification of vulnerable tidal wetlands and inform proactive intervention before irreversible conversion occurs.**

Comment 1.4 The authors should clarify how their independent set of reference points were evaluated and classes determined. The overall accuracy of 99.42% is high and ultimately rather uninformative. While coastal ecosystems are defined it is unclear how a 30 m pixel is assessed, i.e., how were individual pixels evaluated what % of an ecosystem was necessary to evaluate it as that ecosystem. Tidal marsh having the fewest reference points does not make sense giving the goals of the research and what is trying to be conveyed by the accuracy assessment. As currently distributed, the overall accuracy conveys the ability of the classifier to successfully identify other and water classes. Ideally, the accuracy assessment would include change categories as this manuscript is a change analysis.

Response: Thank you very much for these detailed and constructive comments regarding the accuracy assessment. It is done as suggested in the added section “**Accuracy Assessment and bias adjustment**” in the Methods (**Line 503**). This new section explicitly clarifies how to select and interpret the independent validation sample units, and how to use them to assess the accuracy of map and conduct the map area adjustment, following “good practice” recommendations for land cover land change analysis (Olofsson et

al., 2014). This “good practice” has been a standard guidance in remote sensing domain. For example, NASA Land Product Validation Subgroup make it as a basic step for land cover map evaluation.

(1) “The authors should clarify how their independent set of reference points were evaluated and classes determined.” Done with explicit statement (**Lines 539-540**): **The reference sample was interpreted blind to map labels to avoid confirmation bias, on the reference of high-resolution historical images in Google Earth and time series Landsat observations (1985–2024).**

(2) “The overall accuracy of 99.42% is high and ultimately rather uninformative.” We agree that overall accuracy alone can be difficult to interpret for relatively small tidal-wetland classes in landscapes dominated by open water and uplands, as it primarily reflects classification performance across the entire map rather than the accuracy of specific target classes. However, we also report user’s and producer’s accuracies for each class to explicitly quantify their commission and omission errors. We have clarified this in the manuscript (**Lines 514-517**) and detailed confusion matrix (**Supplementary Table S5**): **Producer’s accuracies (reflecting omission error) for tidal marsh, mangrove, and tidal flat are 97.59 ± 2.67%, 91.91 ± 12.23%, and 100.00 ± 0.00%, respectively, while user’s accuracies (reflecting commission error) are 94.59 ± 3.07%, 99.34 ± 1.09%, and 94.08 ± 3.16%, respectively.**

(3) “While coastal ecosystems are defined it is unclear how a 30 m pixel is assessed, i.e., how were individual pixels evaluated what % of an ecosystem was necessary to evaluate it as that ecosystem.” we applied a fractional-cover criterion and gave tidal wetlands priority in labeling. So, if approximately $\geq 20\%$ of a pixel was visually identifiable as tidal wetland, it was labeled as tidal wetland even if other covers were present. The value of 20% followed the rule of NLCD when labelling forests pixels. However, validation points that could not be confidently assigned were excluded and replaced with other randomly selected locations in the same strata to avoid ambiguous class attribution. For non-tidal wetland sample (i.e., open water and uplands), the cover type was otherwise assigned based on dominant class. We provide this information in the manuscript at **Lines 539-547**.

(4) Tidal marsh having the fewest reference points does not make sense giving the goals of the research and what is trying to be conveyed by the accuracy assessment. Validation samples were generated using a stratified random sampling design, where strata were defined by mapped land-cover and land-cover change categories. The number of samples per class was determined by proportional area, with a minimum of 150 samples per class, following established “good practices” that are widely adopted in the field (Olofsson et al., 2014). As tidal marsh occupies a smaller proportion of the total mapped area, it consequently has fewer reference samples. Thus, the weight of each sample varies for each class. So, these sample weights are accounted for in the post-stratified estimator, ensuring unbiased accuracy and area estimates. This is stated in **Lines 509-513**.

(5) As currently distributed, the overall accuracy conveys the ability of the classifier to successfully identify other and water classes. Overall accuracy is largely driven by dominant classes such as open water and uplands, which is why we also emphasize class-specific user’s and producer’s accuracies, and error-adjusted area estimates for each class, including tidal wetlands (refer to **Lines 511-517**).

(6) Ideally, the accuracy assessment would include change categories as this manuscript is a change analysis. We agree that a change-focused study requires explicit validation of change categories. In the revised version, we have added the accuracy assessment for tidal-wetland cover change (i.e., permanent

loss, permanent gain, fluctuations, others), following “good practice” recommendations as well (Lines 527-538).

Reference: Olofsson, Pontus, et al. 2014, Good practices for estimating area and assessing accuracy of land change. *Remote sensing of Environment* 148: 42-57. <https://doi.org/10.1016/j.rse.2014.02.015>

Comment 1.5 The discussion is too focused on the importance of this work and would be strengthened with additional citations surrounding the current policies, proposed policies, modeling, and in situ research that also supports many of the statements in the discussion.

Response: We have revised and extended the Discussion to better situate our findings within existing policy, modeling, and in situ research on tidal wetland dynamics. Please refer to **Discussion Section** for the details (**From Line 255**). Below are some examples of new version of **Discussion**:

(1) Link with modeling-based processing, as (Lines 302-306): **Previous modeling-based approaches to predict tidal marsh resilience primarily considered chronic stressors of accelerated sea level rise and spatial variations in sediment supply. There is ongoing debate about whether the degree of vulnerability is under- or over-estimated for US tidal marshes based on modelling their adaptation potential in horizontal (landward advance) and vertical (upward growth) directions** ^{7,28,41-44}.

(2) Empirical work on restoration and management, as (Lines 331-341): **Multiple regional projects have actively restored tidal wetland ecosystems by removing and updating dike and impoundment systems to re-establish tidal exchange** ⁵⁴⁻⁵⁷. Restoration has also been achieved through managed wetland creation and sediment-based interventions implemented at regional scales ⁵⁸. Large-scale restoration programs in San Francisco Bay ⁵⁹, for example, demonstrate how coordinated marsh creation, sediment enhancement, and phased tidal reintroduction can generate net tidal-wetland gains, illustrating the potential for proactive restoration to counterbalance climate-driven losses when applied at sufficient scale. In parallel, post-disturbance restoration following extreme events should be a critical component of adaptive management. Widespread mangrove dieback after major hurricanes ^{60,61}, and storm-driven marsh degradation⁵⁹ highlight the importance of hydrological rehabilitation and assisted recovery to promote regeneration and reduce the risk of irreversible ecosystem transitions.

(3) Place our observation-based results within the broader scientific and policy context, as (Lines 318-320): **While storms had sometimes been viewed as beneficial to marshes through mineral sediment delivery and associated elevation gains** ^{47,51,52}, this assumption is increasingly challenged under conditions of rising storm frequency and intensity. And (Lines 324-328) **“Reversing the accelerated loss trajectory of US tidal wetlands will require an increase in investment, and a fundamental rethinking of coastal adaptation. Although regulatory protection has successfully limited direct conversion of tidal wetlands, these largely static approaches have proven insufficient to sustain tidal wetlands under accelerating sea-level rise and increasing extreme weather, highlighting the need for adaptive restoration strategies** ^{8,9}.

Comment 1.6 Line 29 – Ecosystem services can be illustrative but comparisons such as these seem reductive and not particularly illustrative of a particular point i.e., instead of using ecosystems services discuss in more detail why tidal wetlands are more valuable to humans than tropical forests?

Response: Done. We have revised the text to clarify why tidal wetlands exhibit higher per-area monetary values than many terrestrial ecosystems, including tropical forests, in global assessments, as **(Lines 28-30): On an area-normalized basis, global assessments rank tidal wetlands as the second most valuable ecosystem after coral reefs, largely because their services are delivered in densely populated coastal zones³, where they directly reduce risk and support human livelihoods^{4,5}.**

Comment 1.7 Line 33 – Peculiar wording. Land use change “changes in land use, resulting in widespread wetland filling.” Wetland filling is the process through which the land use change is happening.

Response: We have revised the sentence, as **(Lines 31-34): For example, since European colonization in the United States (US), extensive conversion of tidal wetlands has occurred, primarily through land-use practices involving wetland filling and drainage associated with agriculture, urbanization, and infrastructure development⁶.**

Comment 1.8 Line 43 – are they truly protected? Protected area vs. the minimal and eroding protections provided by Clean Water Act should be clarified. The protection of these ecosystems is brought up again in the discussion and exactly what is protected and how should be clear from the first mention.

Response: Thank you for this comment. We focus on regulatory policies that limit direct land-use conversion because they operate broadly across the conterminous United States and provide a relatively consistent policy framework over time. This reduces confounding from immediate development pressures and allows us to more clearly assess tidal wetland responses to climate-driven stressors and indirect human influences, and to evaluate whether such regulatory protections are sufficient to sustain these ecosystems under ongoing environmental change.

We acknowledge your point regarding protected and conserved areas (PCAs). Although extensive terrestrial and marine protected areas (<https://www.protectedplanet.net/en>) exist in the United States, assessing PCA impacts on tidal wetlands involves substantial uncertainty and is beyond the scope of this study. Most PCAs were not designed specifically to conserve highly dynamic tidal-wetland systems, and robust, like-for-like comparisons are difficult to establish at national scales. Accordingly, we do not include PCA-based analyses and instead focus on nationwide regulatory policies that limit direct land-use conversion. Nevertheless, we have revised the manuscript to explicitly describe the regulation policy on tidal wetland as below:

In the introduction, we highlight our aim as (Lines 42-50): The adaptability of tidal wetlands to climate change, when direct human alterations and land use change are restricted, remains an enigma. Concurrently, the effectiveness of societal conservation of these ecosystems, achieved through regulatory protection, has been regarded as a topic of significant importance. The US is an ideal setting for investigating tidal wetland resilience and adaptation to climate change. Over the past forty years, the US has implemented regulatory policies specifically designed to

safeguard tidal wetlands by limiting direct land-use conversion ^{7, 13}. Hence, it is intriguing to consider the US as a case study, to investigate the outcomes of extensive tidal wetland protection measures as a counterbalance to the accelerating pace of climate change and the associated degradation of habitats and biodiversity.

In the discussion, we highlight the importance to increase the active restoration, as (Lines 324-328): Reversing the accelerated loss trajectory of US tidal wetlands will require an increase in investment, and a fundamental rethinking of coastal adaptation. Although regulatory protection has successfully limited direct conversion of tidal wetlands, these largely static approaches have proven insufficient to sustain tidal wetlands under accelerating sea-level rise and increasing extreme weather, highlighting the need for adaptive restoration strategies ^{8,9}.

Comment 1.9 Line 60 – Landsat 1 launched in 1972, while there are obvious reasons the analysis does not and is not expected to include the MSS era Landsat data they are nevertheless a part of the Landsat archive and should not be erased by the sentence phrasing.

Response: We agree. We have revised the related descriptions in both the Main and Methodology sections to acknowledge the MSS era of Landsat:

Introduction (Lines 63-65): “We developed a comprehensive monitoring framework that leverages the full 39-year Landsats 4-8 archive (176,223 images) to produce the first annual, 30-m resolution map series of tidal wetland extent and change for the conterminous US (1985-2023).”

Methods (Lines 388-389): We excluded Landsat 1–3 (MSS) data because differences in data resolution, temporal density, and radiometric consistency relative to later sensors limit their suitability for long-term time-series analysis.

Comment 1.10 Line 65 – “...revealing a reality that is far more alarming than previously understood.” seems a little out of touch with the literature both in situ and local to regional remote sensing studies including the author’s own work (Potter 2021; Watson et al. 2017; Vinent et al. 2021; Yang et al. 2024; Di Vittorio et al 2025; Ganju et al. 2025). The writing is suggesting a shocking new finding but in reality, this is additional evidence supporting the previous observations and unifying them behind a robust remote sensing framework.

Response: Thank you. We have revised the sentence to now focus on the methodological and analytical advances provided by our continental-scale, high-frequency time-series analysis, as **(Lines 67-69): This high-frequency time series of the US coastline allows us, for the first time, to move beyond static maps to continuous tracking, enabling continental-scale quantification of rates, acceleration, and shifting drivers of tidal wetland change.**

We also explicitly acknowledge several field-based and local-to-regional studies to contextualize our findings. Specifically, Potter (2021) is cited to relate the impacts of Hurricanes Wilma, Rita, and Katrina on Gulf of Mexico wetlands in 2005 (**Line 153**); Vinent et al. (2021), Watson et al. (2017), and Ganju et al. (2025) are used to support prior evidence that gradual loss of elevation capital increases wetland vulnerability (**Line 347**); Yang et al. (2024) is cited to frame future research directions building on dense

time-series analyses (see **Comment 1.3**); and Di Vittorio et al (2025) is additionally referenced to demonstrate the effectiveness of dense time-series approaches for capturing wetland dynamics (**Line 422**).

Comment 1.11 Line 93 – “... the vital fingerprint of climate change...” unclear at this point in the text how Figure 1. Illustrates the vital fingerprint of climate change or what exactly is being referred to is it temperature, sea level rise, droughts, hurricanes?

Response: We have revised the text to remove this phrasing and instead explicitly describe what is characterized by our continuous analyses, as (**Lines 98-100**): **This high-frequency, continuous analysis enables, for the first time, direct quantification of tidal wetland dynamics associated with climate-related drivers, such as sea-level rise and extreme weather events** (Error! Reference source not found.d-i).

Comment 1.12 Line 107 – “...Coast of Gulf marsh loss” change to “Gulf coast marsh loss” or some other term.

Response: Done. We have changed it to “Gulf coast marsh loss” and have replaced all “Coast of Gulf” by “**Gulf coast**” in the revised manuscript.

Comment 1.13 Line 112 – “...Louisiana estuary...” Unclear what is being referred to here as this is not the name of an estuary as far as I am aware. A particular estuary should be referred to by name.

Response: Done. We have revised all terms of “Louisiana estuary” to “**Mississippi River Delta Plain**”.

Comment 1.14 Line 169 – Greater detail on this impediment to migration including how forest resilience to SLR might be a factor.

Response: Good point. We have revised the explanation with both anthropogenic barriers as well as the forest resilience. Recent regional analyses of coastal forests along the US Atlantic coast (Yeung et al., 2025) demonstrate that tree mortality and forest retreat often lag behind hydrologic change, with forests persisting under chronic salinization and episodic flooding for extended periods before conversion to marsh or shrubland. Such lagged responses can create a bottleneck at the marsh–upland boundary, limiting accommodation space even where topography would otherwise permit migration. Together, forest resilience and anthropogenic constraints likely decouple marsh loss from compensatory inland gains, contributing to net marsh area decline across the elevation gradient.

Please refer to our revision as (**Lines 171-176**): **Our analysis of marsh area across elevation gradients (Fig. 3) reveals that while most loss occurred at low elevations, there were no corresponding significant gains at higher elevations (MK test with 95% confident interval), indicating that lateral migration is not keeping pace with seaward marsh loss. In addition to physical barriers such as roads and developed uplands, forested areas adjacent to marshes may exhibit resistance to early stages of sea-level rise, thereby delaying landward marsh migration and ecological transition** ²⁷⁻³¹.

Comment 1.15 Line 270 – “Our findings suggest that a conservation paradigm focused on static protection and mitigating chronic drivers is no longer sufficient.” without citation it is very unclear who or what the authors are referring to. The authors should identify exactly what policies and practices their research identifies as being inadequate and where.

Response: Done. We revised the sentence and supported it with relevant literature (Campbell et al., 2022; Kirwan and Megonigal, 2013; and He et al., 2025), as **(Lines 324-328): Reversing the accelerated loss trajectory of US tidal wetlands will require an increase in investment, and a fundamental rethinking of coastal adaptation. Although regulatory protection has successfully limited direct conversion of tidal wetlands, these largely static approaches have proven insufficient to sustain tidal wetlands under accelerating sea-level rise and increasing extreme weather, highlighting the need for adaptive restoration strategies** ^{8,9}.

Comment 1.16 Line 280 – The sentence seems to conflate the time series derived abrupt vs. gradual which are spatial and the interpreted classes which are referred to as drivers but not spatially derived. Partially clarified in the subsequent paragraph, it still needs to be clearer in the text.

Response: Thank you. In the revised manuscript, we have removed references to drivers in this sentence and now focus explicitly on the spatial identification of loss hotspots.

The revised text reads **(Lines 341-344): Together, substantially increasing the extent, pace, and strategic targeting of restoration is essential to reverse current loss trajectories and strengthen the long-term resilience of U.S. tidal wetlands, and our identification of loss hotspots offers a practical roadmap for prioritizing restoration actions.**

Comment 1.17 Line 445 – remove “other” as 3DEP is not a global DEM. A better comparison would be DeltaDTM or other global coastal DTM products (Pronk et al. 2024).

Response: Thanks for bringing these data to our attention. We have revised our sentence about 3DEP **(Lines 405-408): “The 3DEP Digital Elevation Model (DEM) product has an absolute vertical accuracy of approximately 0.82 meters root mean square error (RMSE) across the US, which has a higher accuracy than that of widely used global DEM products, such as NASADEM (3.30 m) and Shuttle Radar Topography Mission (SRTM) (3.79 m) (Stoker and Miller, 2022) at a spatial resolution of 30 m.”**

Comment 1.18 Figure 1. The trends texture visualization makes the figure hard to interpret quickly and difficult to see differences in clusters of small estuaries.

Response: Thank you for this helpful suggestion. In the revised figure **(Line 71)**, we simplified the symbology by replacing the dense vertical and horizontal line textures with clearer crossing and horizontal patterns, which improves visual separation and readability.

Comment 1.19

Potter, C., 2021. Remote sensing of wetland area loss and gain in the western Barataria Basin (Louisiana, USA) since Hurricane Katrina. *Journal of Coastal Research*, 37(5), pp.953-963.

Di Vittorio, C.A., Wiles, M., Rabby, Y.W., Movahedi, S., Louie, J., Hezrony, L., Cifuentes, E.C., Hinchman, W. and Schluter, A., 2025. Mapping coastal wetland changes from 1985 to 2022 in the US Atlantic and Gulf Coasts using Landsat time series and national wetland inventories. *Remote Sensing Applications: Society and Environment*, 37, p.101392.

Ganju, N.K., Ackerman, K.V., Defne, Z., Mariotti, G., Curson, D., Posnik, Z., Carr, J.A. and Grand, J., 2025. A Simple Predictive Model for Salt Marsh Internal Deterioration Under Sea-Level Rise and Sediment Deficits: Application to Chesapeake Bay. *Estuaries and Coasts*, 48(6), p.178.

Vinent, O.D., Herbert, E.R., Coleman, D.J., Himmelstein, J.D. and Kirwan, M.L., 2021. Onset of runaway fragmentation of salt marshes. *One Earth*, 4(4), pp.506-516.

Yang, X., Zhu, Z., Kroeger, K.D., Qiu, S., Covington, S., Conrad, J.R. and Zhu, Z., 2024. Tracking mangrove condition changes using dense Landsat time series. *Remote Sensing of Environment*, 315, p.114461.

Pronk, M., Hooijer, A., Eilander, D., Haag, A., de Jong, T., Vousdoukas, M., Vernimmen, R., Ledoux, H. and Eleveld, M., 2024. DeltaDTM: A global coastal digital terrain model. *Scientific Data*, 11(1), p.273.

Response: We have cited these works to support our analysis. Please refer to our response to **Comment 1.10** and **Comment 1.3**.

Reviewer #1 (Remarks on code availability):

Comment 1.20 Previously I reviewed the code. It worked as expected and is well documented.

Response: We thank you for testing our code and for the positive feedback. We are glad to share the code and will continue to maintain and update it.

Reviewer #2 (Remarks to the Author):

Comment 2.1 This is a welcome and important contribution and I unreservedly recommend publication. The potential impact of sea-level rise on coastal wetlands has been a major theme of research in recent years, and good progress has been made understanding the tipping points associated with mangrove and tidal marsh survival. This paper provides a comprehensive analysis of tidal marsh and mangrove habitat change at high temporal and spatial resolution for the conterminous USA. The United States is a fascinating case study, given strong regional differences in the rate of sea-level rise and exposure to extreme storms. The results provide perhaps the first detailed continental-scale assessment of the relative contributions of sea-level rise, extreme storms and anthropogenic stressors in shaping the extent of these important wetlands. The finding that sea-level rise is historically the dominant driver, but that extreme storms are increasing in their relative contribution is a very important finding. As the authors argue in their concluding statements, the two drivers are increasingly interacting in driving a trend of reduced tidal marsh extent. Mangrove low latitude losses appear offset by poleward expansion for now. These findings align with projections of theoretical studies, but it is very exciting (if disturbing) to see clear data demonstrating these patterns.

Response: We sincerely thank you for this very positive and encouraging evaluation. We appreciate your careful reading of the manuscript and your recognition of the importance of our findings on the interacting roles of chronic stressors and acute shocks in shaping shifting tidal wetland dynamics across the conterminous United States. Please see below for our point-by-point response to the review comments.

Comment 2.2 One suggestion that might not be too onerous. The findings give rise to the question of whether the increased losses to storms are due to increased exposure (i.e. an increase in the frequency and intensity of storms) or increased vulnerability (lower tidal frame due to sea-level rise), or both. One way of considering this is to look at data relating to storm frequency/intensity trends in these regions. Might this be considered?

Response: Thank you very much for this excellent suggestion. We fully agree with the distinction raised: increased losses could stem from increased exposure (e.g., more frequent/intense storms) or increased vulnerability (e.g., loss of elevation capital due to sea-level rise).

In this study, we rely on dense Earth observation time series to identify the proximate drivers of change, that is, the immediate triggers (like a specific hurricane) that cause cover conversion. This allows us to constrain when loss occurs. However, disentangling the underlying mechanisms, such as determining whether a loss event occurred because the storm was historically intense (exposure) or because the marsh had gradually lost resilience (vulnerability), requires integrating complex hydrodynamic modeling with long-term meteorological trends.

We are actively exploring this specific “driver regime shift” question in a separate, ongoing study. We believe this attribution cannot be robustly addressed within the remote-sensing framework of the current manuscript, which is designed to identify the direct agents of change.

We have therefore **revised the Discussion (Lines 345-352)** to explicitly acknowledge this distinction as: **While our large-scale analysis identifies the dominant proximate drivers of change (i.e., the immediate triggers of loss), disentangling the specific contributions of increased exposure (e.g., shifts in storm intensity regimes ^{35,63,64}) versus increased vulnerability (e.g., gradual loss of elevation capital ⁶⁵⁻⁶⁸) remains a challenge. We hypothesize that the observed acceleration is likely the result of synergistic interactions between these factors: chronic stressors condition the landscape to be more vulnerable to acute shocks. However, separating their individual roles requires integrating complex meteorological data with localized marsh health metrics. Future work coupling this synoptic remote sensing record with process-based models has the potential to disentangle mechanisms ⁶⁹.**

Comment 2.3 Secondly, I was a little surprised to find that several recently published observational and theoretical papers published in leading journals (e.g. papers in *Science*, and *Nature* since 2020) providing context for these observations were not cited. A quick Google Scholar search will identify these. Some of the referencing seems a little dated.

Response: Thanks for your suggestion. We have carefully revisited recent publications to ensure that the discussion is well grounded in the most current understanding of tidal wetland dynamics.

As a result, we have added several recent papers published since 2020 (Ensign et al., 2023, *Science*; Tognin et al., 2021, *Nature Geoscience*; Yeung et al., 2025, *Nature Sustainability*; He et al., 2025, *Nature Review Biodiversity*; Cornwall, 2025, *Science*). These additional citations support the sections on climate-driven coastal habitat change (**line 176**), extreme-event impacts (**line 319**), tidal marsh vulnerability, constraints on wetland conservation (**line 288**), and regional restoration projects (**Line 334**).

Beyond these two suggestions, my observations are of a minor nature, as follows:

Comment 2.4 Line 58: “Here, we provide the definitive, observation-based answer to this question”. Make it clearer what this question is.

Response: Done. We re-wrote the sentences to explicitly point out the gap and question (**Lines 58-63**): **These methods are largely blind to the crucial, inter-annual rates of change and acceleration that define an ecosystem's true trajectory and vulnerability, leaving a critical gap in our understanding of whether decades-old conservation policies remain effective under accelerating sea-level rise and increasing climate volatility. Here, we provide a definitive, observation-based assessment of long-term tidal wetland trajectories across the conterminous US.**

Comment 2.5 Line 80 (figure caption). There is no such place as “Louisiana estuary”. Perhaps “estuaries of Louisiana?”

Response: Fixed. We revised all terms of “Louisiana estuary” to “**Mississippi River Delta Plain**”.

Comment 2.6 Line 81. Instead of “seaward erosion” I would suggest “seaward edge erosion”. The former gives the sense that the marsh is eroding in a seaward direction.

Response: Thank you and revised accordingly (**Line 83**).

Comment 2.7 Line 82. It would be good to give some coordinates for this figure

Response: Done. We have added geographic coordinates for all case study areas (such as **San Francisco Bay (38.14° N, 122.29° W)**) in the caption of Fig. 1 to improve spatial clarity (**Lines 81-88**).

Comment 2.8 Line 89. I suggest you cite the loss here as negative (-19,931) in common with the other estimates

Response: The value reported here (19,931 km²) refers to the absolute estimated tidal-wetland area in 2023 (i.e., “**In 2023, the total estimated area of tidal wetlands in the conterminous US was 19,931 [±1,046] km², 95% confidence interval**”) and is therefore expressed as a positive value. To ensure consistency and avoid confusion, we have thoroughly reviewed the manuscript to ensure all loss estimates are consistently reported as negative values.

Comment 2.9 Line 93. ...year-2. Shouldn't this just be year-1?

Response: The unit yr⁻² is correct here because the metric represents an acceleration rate, indicating the rate of change in annual tidal-marsh loss over time. It quantifies how much the annual loss rate itself increases (or decreases) per year, rather than the loss rate (with unit yr⁻¹).

Comment 2.10 Line 124. There is an apparent contradiction in the wording here: “...show only marginal declines in loss rates but are marked by accelerating trajectories of loss”. How is the rate different to the trajectory?

Response: In our terminology, the loss rate refers to the magnitude of annual tidal-marsh loss (i.e., the speed of loss), whereas the trajectory describes how that rate changes over time (i.e., whether the loss rate is accelerating or decelerating). We have revised the sentence to make it clear, as (**Lines 127-129**): **In contrast, sections of the Atlantic Coast, where the long-term resilience of tidal wetlands remains uncertain, show comparatively low absolute rates of loss but are marked by accelerating trajectories of loss, underscoring heightened vulnerability.**

Comment 2.11 Line 211. You don't mention extreme freezing events as a driver of mangrove dynamics in the Gulf. This seems odd given the voluminous literature on this issue.

Response: We agree that extreme freezing events are an important driver of mangrove dynamics in the Gulf region. Severe freeze events in Florida during the winters of 1989–1990 and 2010 caused widespread mangrove dieback and obviously detected in this study (two peaks of mangrove diebacks in Fig. 2b).

In the revised manuscript, we explicitly incorporate freeze events into the discussion of mangrove dynamics to better reflect their role as a key form of extreme disturbance, as (**Lines 217-221**): **Our analysis reveals a critical increase in the occurrence of large-scale mangrove diebacks following major hurricanes and severe freeze winter, particularly in the historical core of their range in south Florida. Hurricane-related dieback (notably in 2005 and 2017) is concentrated along seaward mangrove margins, whereas dieback following extreme freeze events (notably in 1990 and 2010) occurs predominantly along upland boundaries.**

Comment 2.12 Line 273. “The notion that storms can be beneficial to marshes must be re-evaluated...”. This seems to need a reference to a paper making this claim.

Response: The literature reports both positive and negative impacts of storms on marsh resilience, depending on storm characteristics and environmental context. We have revised the sentence to explicitly acknowledge this debate with added appropriate citations, as **(Lines 318-320): While storms have sometimes been viewed as beneficial to marshes through mineral sediment delivery and associated elevation gains^{47,51,529}, this assumption is increasingly challenged under conditions of rising storm frequency and intensity.**

Comment 2.13 Line 278. You mention the importance of identifying and supporting corridors for landward expansion here, a point often made. However, do your results support the contention that these corridors are actually being used in any widespread way?

Response: We appreciate your valuable comments. Our analysis does not quantify the extent of widespread landward migration of tidal wetlands at the US scale. Although our maps do capture localized landward expansion, such as mangrove encroachment (Fig. 1i) and marsh migration to upland forests (Supplementary Fig. S9c), we do not assess how broadly such migration corridors are currently being used nationwide.

We have revised the text to avoid implying widespread corridor use and instead frame landward migration as a potential conservation strategy, supported by prior studies. Please refer to revised version as **(Lines 328–333): First, the large areal historic conversions of tidal wetland since European colonization in the US⁶, including an estimated ~4,800 km² of restricted and impounded wetlands⁵³ and ~2,400 km² of drained former wetlands²⁷, represents a major restoration opportunity, particularly where tidal connectivity can be re-established³⁰. Multiple regional projects have actively restored tidal wetland ecosystems by removing and updating dike and impoundment systems to re-establish tidal exchange^{54–57}.**

Comment 2.14 Line 589. The photos in (b) are a little too grainy in my version

Response: The images were derived from historical satellite imagery available in Google Earth, which can vary considerably in quality. We replaced the grainy photo with a clear one **(Supplementary S-8b)**.

Comment 2.15 Line 590. Spelling “Hurricane Katrina that led to..” not “lead”

Response: Fixed.

Reviewer #3 (Remarks to the Author):

Comment 3.1 It was a pleasure to review the manuscript by Yang et al., which provides an insight into the change in saltmarsh extent in the USA. I found the manuscript to be interesting and generally well-written. However, the manuscript requires extensive revisions before it can be considered for publication. Currently, the manuscript has too much focus on the USA. As a result, it is at risk of being too parochial and lacking wider appeal. For example, I would recommend the authors incorporate further international examples into the introduction to increase the global application and relevance of the work.

Response: Thank you for this valuable suggestion. In the revised manuscript, we now begin the **Introduction** by highlighting global patterns of tidal wetland loss and degradation, drawing on recent international studies to establish the worldwide relevance of the processes examined here. We then frame the conterminous United States as a focused case study, motivated by its extensive tidal wetlands and long-standing regulatory limits on direct land-use conversion. This structure clarifies that our US-based analysis addresses globally relevant questions about tidal wetland resilience under climate change in settings where direct land use conversion is constrained.

Please refer to **(Lines 40-50): Despite growing recognition of their ecological and climatic importance, tidal wetlands still experienced substantial global declines in extent and condition over recent decades, primarily driven by intense human activities and increasing climate-related stressors^{1,8,11,12}. The adaptability of tidal wetlands to climate change, when direct human alterations and land use change are restricted, remains an enigma. Concurrently, the effectiveness of societal conservation of these ecosystems, achieved through regulatory protection, has been regarded as a topic of significant importance. The US is an ideal setting for investigating tidal wetland resilience and adaptation to climate change. Over the past forty years, the US has implemented regulatory policies specifically designed to safeguard tidal wetlands by limiting direct land-use conversion^{7,13}. Hence, it is intriguing to consider the US as a case study, to investigate the outcomes of extensive tidal wetland protection measures as a counterbalance to the accelerating pace of climate change and the associated degradation of habitats and biodiversity.**

Comment 3.2 The novelty of study has been other emphasised. We know that marshes are not stable and other studies have investigated the change in extent of saltmarshes both in the USA and elsewhere. The novelty comes from how the authors have considered tidal variability and it is important that the authors do not “overplay” the originality of the work.

Response: We appreciate your question about our progress compared with other tidal wetland change analyses. We agree that tidal wetlands are well known to be dynamic and that previous studies have quantified their changes in the US (e.g., CCAP-based analyses by Osland et al., 2022 and Crooks et al., 2018) and globally (such as Murray et al 2022 and Campbell et al., 2022). These products rely on using sparse satellite imagery over multi-year epochs to develop maps every few years. As a result, they are limited in their ability to yield fine-scale loss or gain estimates, estimates of change rates for sub-annual periods, and fine scale estimates of the timing of change occurrences.

In our study, we addressed tidal variability to continuously track tidal wetland changes. This dense time-series framework enables **two advances** that are difficult to achieve with traditional epoch-based approaches:

(1) Quantification of **acceleration** in tidal wetland loss, made possible by annual maps that allow moving-window analyses of changing loss rates.

(2) Attribution of dominant drivers, enabled by the **timing of cover conversion** and distinguishing gradual change under chronic stressors from abrupt change associated with acute events.

We note these advances and have clarified the revision with the changes below:

(1) We added a schematic illustration (**Supplementary Fig. S-14, Page 27**) demonstrating why dense time-series monitoring provides additional insight into gradual versus abrupt change processes compared with epoch-based mapping.

Figure S-1. Conceptual comparison of epoch-based and dense time-series approaches for detecting tidal wetland change. a, Epoch-based mapping compares land cover at two disconnected time slices and identifies net change, but provides limited information on the timing or underlying processes driving that change. b,c, Dense time-series approach tracks wetland condition continuously through time, enabling differentiation between gradual change driven by chronic stressors (b) and abrupt change triggered by acute pulse events (c). See Fig. S8–S10 for real Landsat time-series examples of gains and losses associated with chronic and acute drivers.

(2) We revised the **Introduction** to explicitly describe the limitations of traditional epoch-based assessments and to frame our contribution by advancing continuous monitoring rather than redefining the existence of change, as (Lines 51-63): **Historically, tidal wetland extent has been difficult to evaluate synoptically due to the inherent dynamic and heterogeneous nature of coastal systems**¹. Water level fluctuations lead to the varying inundations of tidal wetlands over time and inconsistency of the spectral values in the satellite images (Supplementary Figure S-1). This variability has been a significant barrier for continuous monitoring of tidal wetlands. Traditionally, optimal satellite images, such as those captured during similar high tide or low tide conditions, were selected based on tide modeling and measurements from gauge stations^{12,14,15}. Previous large-scale assessments have been limited to infrequent, epoch-based time slices that under-sample the full variability of these dynamic systems^{7,8,16}. These methods are largely blind to the crucial, inter-annual rates of change and acceleration that define an ecosystem's true trajectory and vulnerability, leaving a critical gap in our understanding of whether decades-old conservation policies remain effective under accelerating sea-level rise and increasing climate volatility.

Here, we provide a definitive, observation-based assessment of long-term tidal wetland trajectories across the conterminous US.

(3) We expanded the **Discussion** to directly **compare** our results with existing products (Lines 259-277) and to emphasize the new insights gained from resolving **acceleration** (Lines 278-289) and **timing** (Lines 290-300).

Comment 3.3 The discussion requires considerable editing, especially as it currently lacks wider context. This mainly comes from the lack of citations in the discussion, but further consideration of the wider application and implications of the work would also be beneficial.

Response: We appreciate this important comment and have substantially revised and expanded the **Discussion (From Line 255)** to strengthen its broader context and relevance. We also added relevant literature to better situate our findings within previous work. Specifically,

(1) Place our findings explicitly within the context of other popular remote sensing-based tidal wetland changes (such as Murray et al., 2022; Campbell et al., 2022; Crooks et al., 2018; and Bunting et al., 2022) (Lines 259-277)

(2) Highlight the unique contribution of dense time-series observations for quantifying acceleration in loss trend and shifting drivers, linking with prior studies (such as Lake 2013; Ohenhen et al., 2023 and He et al., 2025) (Lines 278-300)

(3) Connect our empirical results to ongoing debates from process-based resilience and vulnerability models (such as Schuerch et al., 2018; Törnqvist et al., 2021; Esign et al., 2023; Leonardi et al., 2016) (Lines 301-323)

(4) Discuss implications for restoration, management (such as Cornwall, 2025; Castillo et al., 2022; Lewis et al., 2015; Patrick et al., 2022; Vincent et al., 2021) (Lines 324-344), and future research directions (Cai et al., 2025; Kirwan and Gedan, 2019; Scheffer et al., 2009; Verbesselt et al., 2016; Runion et al., 2025) (Lines 345-366)

Reviewer #1 (Remarks to the Author):

The authors have addressed my previous comments comprehensively. I recommend the manuscript for publication.

Reviewer #1 (Remarks on code availability):

I previously reviewed the code.

Response: We sincerely appreciate your constructive feedback throughout the review process, as well as your thoughtful evaluation of our code.

Reviewer #2 (Remarks to the Author):

The authors have undertaken a comprehensive review of the manuscript in my view adequately addressing the issues raised by each of the reviewers. As suggested in my original review, the work is novel and important, and deserving of publication now that these concerns have been addressed. Could the authors please check line 64- should this read Landsat or Landsats? (the singular?)

Response: We sincerely appreciate your constructive feedback throughout the review process. And we are grateful for your positive assessment of the manuscript and for recognizing its contribution. We have corrected Line 64 and other related locations to read “Landsat” (singular).

Reviewer #3 (Remarks to the Author):

Following the revisions made after the first review, the manuscript by Yang et al is much improved. It is my view that the manuscript should be considered for publication, but only after the reviews re-visit the global context of their work. I acknowledge the amends made by the authors following my comment in the first review regarding the focus on US tidal wetlands but think they could go further contextualise and quantify global loss. For example, recent reports such as Brook et al. (2025) have attempted to quantify global saltmarsh loss (<https://www.wwf.org.uk/sites/default/files/2025-06/state-of-the-worlds-saltmarshes-2025.pdf>). I would like to see the authors incorporate some of these estimates into their introduction to ensure there is global context. The same is true for the abstract, which provides little awareness of the ‘bigger picture’ ie tidal wetland loss on a global scale.

Response: We sincerely thank you for your constructive suggestion and for the positive evaluation of our revised manuscript. We have further strengthened the global context in both the Introduction and the Abstract. Specifically, we now explicitly quantify global tidal wetland losses in the early 21st century, incorporating estimates from recent global syntheses, including **~4,000 km² of net tidal wetland loss (Murray et al., *Science*, 2022) and ~1,400 km² of tidal marsh loss (Campbell et al., *Nature*, 2022; Brook et al., 2025) (Line 43)**. These additions situate our US-based analysis within the broader global trajectory of tidal wetland decline. We have also revised the opening sentence of the Abstract to foreground the global perspective, as **“Tidal wetlands are critical ecosystems for coastal sustainability, yet despite growing regulatory protection, they continue to decline globally” (Line 11)**.